# Active case-finding policy development, implementation and scale-up in high-burden countries: A mixed-methods survey with National Tuberculosis Programme managers and document review

Olivia Biermann[1]*, Phuong Bich Tran[1], Kerri Viney[1,2], Maxine Caws[3,4], Knut Lönnroth[1], Kristi Sidney Annerstedt[1]

1 Department of Global Public Health, Karolinska Institutet, Stockholm, Sweden, 2 Research School of Population Health, College of Health and Medicine, Australian National University, Canberra, Australia, 3 Department of Clinical Sciences, Liverpool School of Tropical Medicine, Liverpool, United Kingdom, 4 Birat Nepal Medical Trust, Lazimpat, Kathmandu, Nepal

* olivia.biermann@ki.se

## Abstract

### Background

The World Health Organization (WHO) stresses the importance of active case-finding (ACF) for early detection of tuberculosis (TB), especially in the 30 high-burden countries that account for almost 90% of cases globally.

### Objective

To describe the attitudes of National TB Programme (NTP) managers related to ACF policy development, implementation and scale-up in the 30 high-burden countries, and to review national TB strategic plans.

### Methods

This was a mixed-methods study with an embedded design: A cross-sectional survey with NTP managers yielded quantitative and qualitative data. A review of national TB strategic plans complemented the results. All data were analyzed in parallel and merged in the interpretation of the findings.

### Results

23 of the 30 NTP managers (77%) participated in the survey and 22 (73%) national TB strategic plans were reviewed. NTP managers considered managers in districts and regions key stakeholders for both ACF policy development and implementation. Different types of evidence were used to inform ACF policy, while there was a particular demand for local evidence. The NSPs reflected the NTP managers' unanimous agreement on the need for ACF scale-up, but not all included explicit aims and targets related to ACF. The NTP managers

**Data Availability Statement:** This is a mixed-methods study based on a sample of National Tuberculosis Programme managers from high tuberculosis burden countries. The respondents are well-known in the fields of tuberculosis and active case-finding; making the full data set publicly available would breach their privacy. The informed consent that all respondents signed promised full anonymity. The raw data set is included as a Supporting Information file without identifying information, qualitative data or indicators that were not used in the analysis. Following data requests, survey transcripts will be reviewed for any potential identifying information and will only be made available to researchers who sign a data sharing agreement. Data requests may be sent to maike.winters@ki.se.

**Funding:** This work was supported by the European Union-Horizon 2020-funded IMPACT-TB project (grant 733174). Funder website: https://ec.europa.eu/programmes/horizon2020/en. The grant was received by MC. The funders had no role in study design, data collection and analysis, decision to publish, or preparation of the manuscript.

**Competing interests:** The authors have declared that no competing interests exist.

**Abbreviations:** ACF, Active tuberculosis case-finding; NSP, National TB Strategic Plan; NTP, National Tuberculosis Programme; TB, Tuberculosis; WHO, World Health Organization.

recognized that ACF may decrease health systems costs in the long-term, while acknowledging the risk for increased health system costs in the short-term. About 90% of the NTP managers declared that financial and human resources were currently lacking, while they also elaborated on strategies to overcome resource constraints.

## Conclusion

NTP managers stated that ACF should be scaled up but reported resource constraints. Strategies to increase resources exist but may not yet have been fully implemented, e.g. generating local evidence including from operational research for advocacy. Managers in districts and regions were identified as key stakeholders whose involvement could help improve ACF policy development, implementation and scale-up.

## Introduction

Tuberculosis (TB) remains one of the top 10 causes of death worldwide [1]. The World Health Organization (WHO) End TB Strategy [2] and the United Nations (UN) Sustainable Development Goals [3] aim at ending TB by 2030. International attention has also been drawn to TB with the 2017 Global Ministerial Conference on Ending TB [4] and the UN's 2018 General Assembly high-level meeting on TB [5]. Due to a combination of underreporting of detected cases and underdiagnosis, there is still a gap of three million between estimated incident TB cases and those notified worldwide [6].

Ending TB will require increased and early case detection [2] by applying strategies such as systematic screening, i.e. the "systematic identification of people with suspected active TB in a predetermined target group, using tests, examinations or other procedures that can be applied rapidly" [7]. Active case-finding (ACF) is tantamount to systematic screening for active TB, though it implies screening outside of health facilities [7]. The potential benefits and risks of ACF need to be carefully balanced when developing and implementing ACF policies [7–11].

This study focuses on the 30 high TB burden countries, as they account for 85–89% of the global TB burden [12]. Moreover, this study focuses on National TB Programme (NTP) managers, key stakeholders in developing and implementing ACF policies [13]. NTPs are usually housed within national ministries of health, though the activities of the Programme are implemented at different levels of the health system [14]. Depending on the context, NTPs are responsible for delivering high-quality and effective diagnostic, treatment and preventive services [13, 14].

The aim of this study was to describe attitudes of NTP managers related to ACF policy development, implementation and scale-up in the 30 high-burden countries, which potentially impact the development and implementation of national ACF policies.

## Materials and methods

### Design

This was a mixed-methods study with an embedded design (Fig 1) [15], complemented by a document review. The study comprised a cross-sectional survey with NTP managers from 30 high TB burden countries, which included closed and open-ended questions designed to elicit quantitative and qualitative information, enhancing each other [15]. The document review included a sample of national TB strategic plans (NSPs) from the high TB burden countries

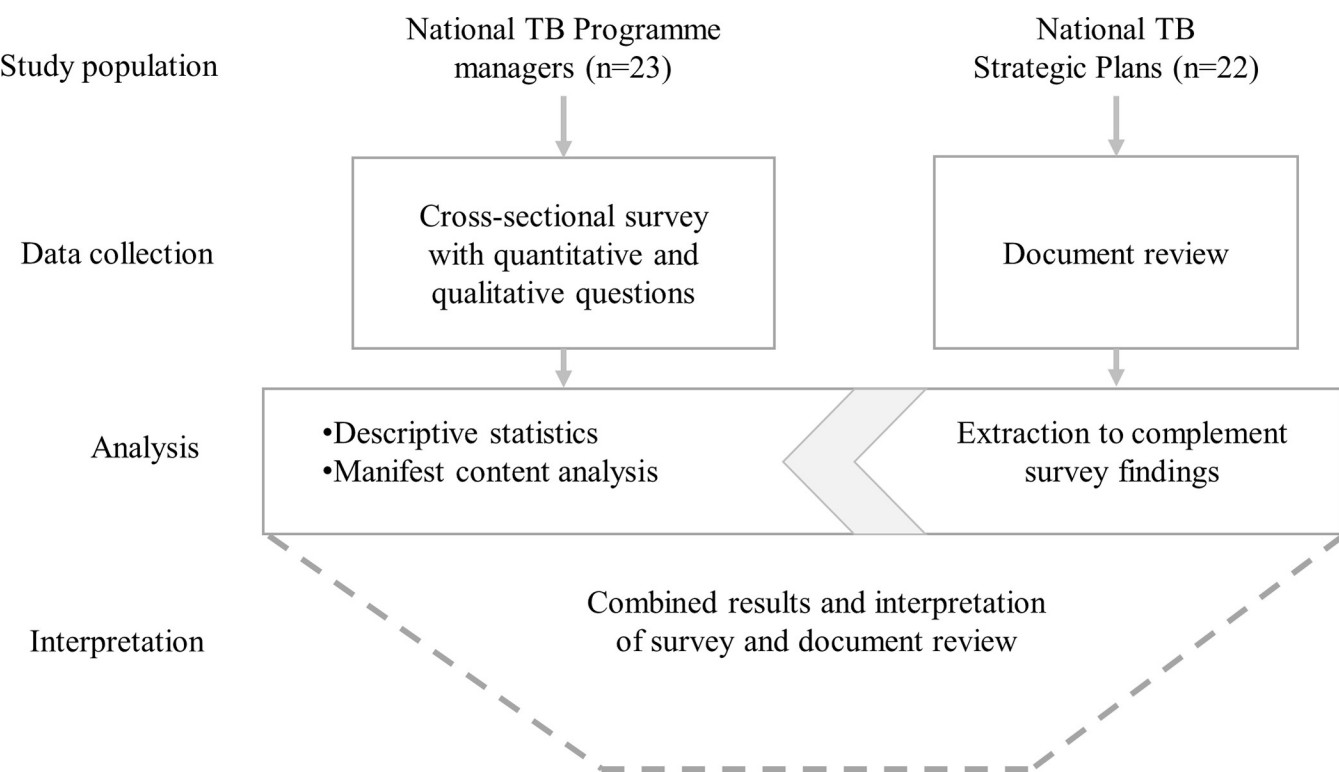

**Fig 1. Pictorial representation of the data collection steps, analysis and interpretation.** 23 National Tuberculosis (TB) Programme managers participated in the cross-sectional survey, which was analyzed using descriptive statistics and manifest content analysis. 22 National TB Strategic Plans were reviewed, and data extracted to complement the survey findings. All data were analyzed in parallel and merged in the interpretation of the findings.

and was designed to complement the findings from the interviews. We implemented all surveys through structured interviews and therefore refer to them as "interviews" in the subsequent text.

## Conceptual framework

The policy analysis triangle [16] considers how the content of a policy, actors, context and processes shape policymaking. We used the triangle as a conceptual framework, adapting it by putting "context", "processes" and "actors" as the corner stones of the triangle (instead of having "actors" in the middle of the triangle, as in the original version) and including example topics we covered (Fig 2). The framework shapes the presentation of the study findings.

## Survey

**Questionnaire.** The development of the questionnaire was informed by a scoping review and an expert interview study on factors influencing the development and implementation of ACF policies [11, 17]. The preceding work considered ACF globally, while this survey focuses on the 30 high TB burden countries. Two independent researchers with expertise in survey design provided feedback on the survey to minimize biases [18].

The questionnaire (S2 Appendix) included sections on the general views on ACF, national ACF policies, evidence use, contextual factors, scale-up, monitoring and evaluation, and lessons learned. Question formats included Likert scales, lists, yes/no questions and open-ended questions (without probing questions). Five-point Likert scales investigated 1) agreement with

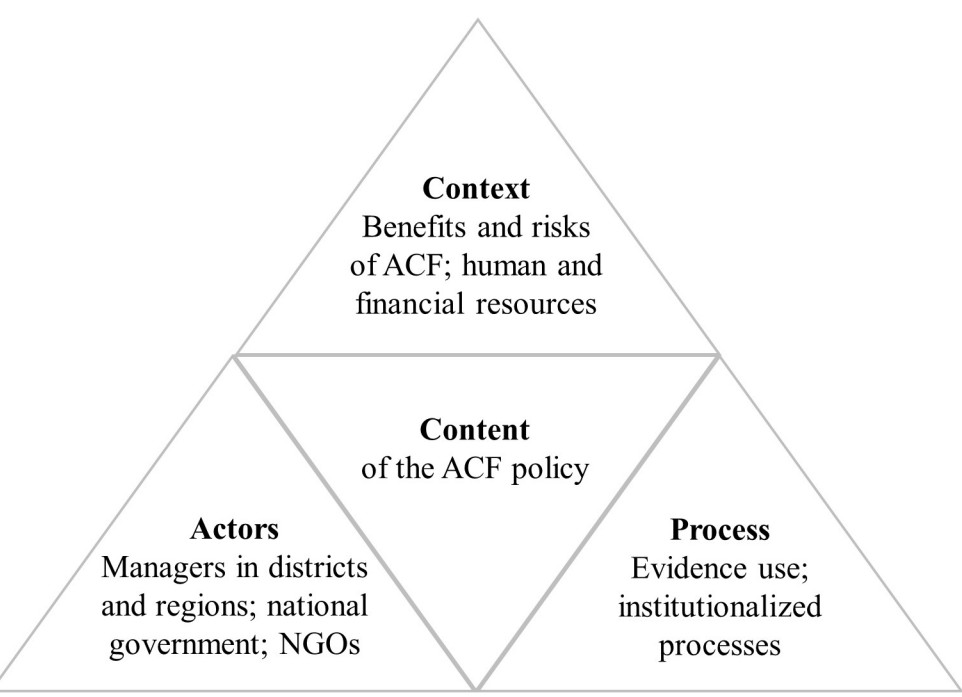

**Fig 2. Policy analysis triangle (adapted from [16]).** This study showed that the content of active case-finding (ACF) policies, including their development, implementation and scale-up, is influenced by context, actors and processes. Example topics: Context–Important factors included the perceptions of the benefits and risks of ACF and the availability of resources. Actors–Key actors comprised the national government, managers in districts and regions and non-governmental organizations (NGOs). Process–Important processes were evidence use and institutionalized processes to facilitate evidence use, e.g. regularly convening working groups.

statements about the potential benefits and risks of ACF, 2) degree of influence of contextual factors on ACF policies, and 3) frequency of use of different types of evidence in ACF policy. The NTP managers also confirmed which stakeholders were involved in ACF policy development and implementation, using a list of 14 types of stakeholders.

**Participants.** NTP managers from all 30 high-burden countries were contacted via email. In total, 23 NTP managers agreed to be interviewed (participation rate: 77%). 17 NTP managers took part in the interview themselves, whereas six appointed one of their team members to participate on their behalf. In this article, we refer to "the NTP managers", meaning all 23 respondents. The NTP managers did not receive an incentive for participation.

NTP managers from the following 23 countries are represented: Bangladesh, Brazil, Cambodia, China, Congo, Democratic Republic of the Congo, Ethiopia, Indonesia, Kenya, Lesotho, Mozambique, Myanmar, Namibia, Nigeria, Pakistan, Papua New Guinea, Philippines, South Africa, Tanzania, Thailand, Vietnam, Zambia and Zimbabwe. NTP managers from seven countries did not participate in the study: Angola, Central African Republic, Democratic People's Republic of Korea, India, Liberia, Russia and Sierra Leone. Five were not interested in or did not have time for being interviewed and one did not reply. One NTP manager started but discontinued the interview due to an unstable telephone connection; several attempts were made to complete the survey but remained unsuccessful due to limited interest and/or time.

**Data collection and management.** The interviews were conducted in English between March and October 2018. As the pilot interview took 54 minutes, we condensed the survey. We included the pilot results in our dataset, apart from the questions that we deleted in the revised questionnaire. The remaining interviews took on average 48 minutes (28–68 minutes).

The primary investigator (OB) conducted ten interviews in person during international conferences, five at a WHO meeting, and eight on the phone/Skype, depending on the participants' preferences. All NTP managers were provided with information about the study and the questionnaire when accepting to participate. After providing informed consent, the interviews were voice-recorded. OB or PT transcribed the responses verbatim and shared the transcriptions with the NTP managers (and appointed team members) for their information and review. Five NTP managers (22%) replied after having received the filled-in survey; one included additional information. If the NTP managers did not respond within four weeks, the survey was considered complete. One interview was terminated half-way due to limited time; we still included the data. PT entered all responses into REDCap (Research Electronic Data Capture), a secure, web-based platform [19]. The anonymity and confidentiality of the NTP managers were ensured by removing all identifiers in the presentation of the results.

**Data analysis.** Using STATA 15 (StataCorp LLC), we computed descriptive statistics (frequencies, mean, median and proportions). We used Cronbach's Alpha to assess the internal consistency of the Likert scale items as background analysis when developing the instrument [20]. We added additional indicators to investigate any patterns in the responses: 1) country income level and 2) region [21], and 3) proportion of NTP budget consisting of domestic funding, 4) international funding and 5) being unfunded [6].

We analyzed open-ended answers with NVivo 11 (QSR International) using content analysis [22], focusing on the manifest content and generating meaning units, codes and categories (Table 1). In the results section, we first describe quantitative then qualitative findings and add information from the document review where applicable. The Checklist for Reporting Results for Internet E-Surveys (CHERRIES) was applied for reporting [23].

## Document review

We identified 18 NSPs through an online search, whilst an additional four NSPs were shared with us by staff from the Stop TB Partnership or WHO. NSPs from the following 22 countries are included: Angola, Bangladesh, Brazil, Cambodia, Ethiopia, India, Indonesia, Kenya, Liberia, Mozambique, Myanmar, Namibia, Nigeria, Pakistan, Papua New Guinea, Philippines, Sierra Leone, South Africa, Tanzania, Vietnam, Zambia and Zimbabwe. The NSPs from Angola, Brazil and Mozambique were available in Portuguese. Eighteen of these countries overlap with the countries that are represented in the survey (the ones that do not overlap are Angola, India, Liberia and Sierra Leone). We were not able to obtain NSPs from the following eight countries: China, Congo, Central African Republic, Democratic People's Republic of Korea, Democratic Republic of Congo, Lesotho, Russia and Thailand. Congo and Lesotho do not have an NSP according to the interviews performed. China, Democratic Republic of the Congo and Thailand have NSPs based on the interviews, but we were unable to gain access. Whether the Democratic People's Republic of Korea, the Democratic Republic of Congo and Russia have NSPs or whether the NSPs are not publicly available remained unclear.

**Table 1. Example of content analysis.**

| Question: What do you consider the most powerful influence in implementing the ACF policy? | | |
|---|---|---|
| Meaning unit | Code | Category |
| "It's the health system context with limited human and financial resources that could negatively influence ACF implementation." (NTP manager 23) | Human and financial resources | |
| "To a large extent, we depend on the collaboration with the local health system; if they don't buy in then that's a failure." (NTP manager 13) | Subnational health system | Health system |

ACF, active case-finding; NTP, National Tuberculosis Programme.

**Data extraction from the national strategic plans.** We developed a data extraction table, which we filled with data on the development of the NSP, background and aims, targets, operational and ethical considerations, budget estimates and funding sources (S1 Appendix). A native Portuguese speaker extracted and translated the data from the NSPs from Angola, Brazil and Mozambique.

**Ethics approval and consent to participate and for publication.** This study has been approved by the Swedish Ethical Review Authority in Stockholm (2017/2281-31/2).

## Results

### Characteristics of NTP managers

Of the 23 NTP managers who responded to the survey, 39% were female (n = 9). Seventy-four percent (n = 17) of the NTP managers identified themselves as policymakers, whereas 13% (n = 3) identified themselves as researchers or "other". Among the 20 participants who identified themselves either as policymakers or "other", 90% (n = 18) reported having research experience. In each of the following sections, we state the total number of participants who answered a particular question, as not all questions were answered by all 23 participants.

### Context

**The perceived benefits of ACF.** All NTP managers (n = 23) asserted that ACF has some benefits (Fig 3). They agreed with the sentiments that ACF leads to early detection, diagnosis and treatment, and reduced TB transmission and incidence. One NTP manager (4%) from a lower middle-income country disagreed that ACF leads to improved treatment outcomes, while another NTP manager (4%) from a low-income country disagreed that ACF leads to reduced future health system cost, and positive social and economic consequences for a TB patient.

Other potential benefits of ACF included better health system performance (e.g. increased access and care-seeking, increased human resource capacity and synergies), epidemiological impact (e.g. reduced TB mortality) and health educational benefits through increased knowledge and awareness.

"Improving knowledge about TB among patients and the community I think is the most important component of the programme." (NTP manager 22)

**The perceived risks of ACF.** The NTP managers' views on potential risks of ACF were more heterogenous compared to their views on the potential benefits (Fig 4). More than half of the NTP managers (55%, n = 12/22) neither agreed nor disagreed that ACF leads to increased false-positive diagnoses of TB; they emphasized that the risk of a false-positive diagnosis depends on the diagnostic test used. Four of the five upper middle-income countries (80%) included in this study were among those who neither agreed nor disagreed. Moreover, while 83% (n = 19/23) of the NTP managers affirmed that ACF leads to increased health system costs in the short term, 83% disagreed that ACF would lead to increased health system costs in the long term (over 10 years); both estimates included NTP managers from low-, lower middle- and upper middle-income countries. The NTP managers brought up two examples of how ACF could impair health system performance; by overburdening health care workers and increasing waiting times for non-TB patients.

**ACF policies in high-burden countries.** According to the NTP managers, a written ACF policy exists in 91% (n = 21/23) of the countries, typically as part of an NSP. In line with that, the majority (87%, n = 20/23) said that ACF contributes to the achievement of the goals

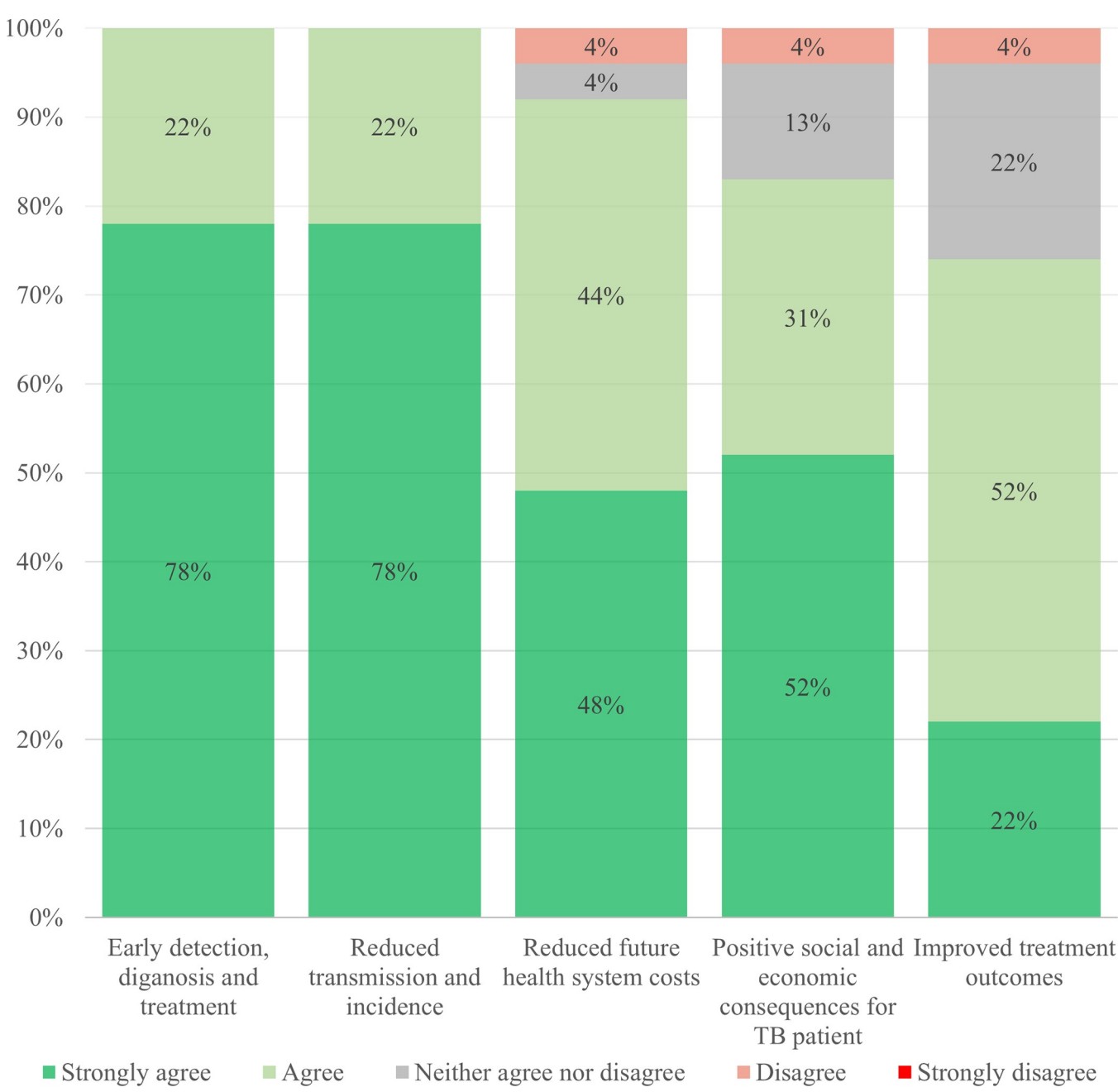

**Fig 3. Perceived benefits of active case-finding.**

outlined in their NSP. In 43% (n = 9/21) of the countries, the national ACF policy has been evaluated or formally assessed.

The document review supported this and showed that 86% (n = 19/22) of the NSPs stated aims directly related to ACF. Another 86% reported a case detection gap. All NSPs explicitly mentioned the need to undertake contact tracing and described one or more additional target groups for ACF. Yet, only six (27%) NSPs outlined concrete targets for the number of people screened and three (14%) (two lower middle-income countries and one low-income country) specified targets for the number of people tested.

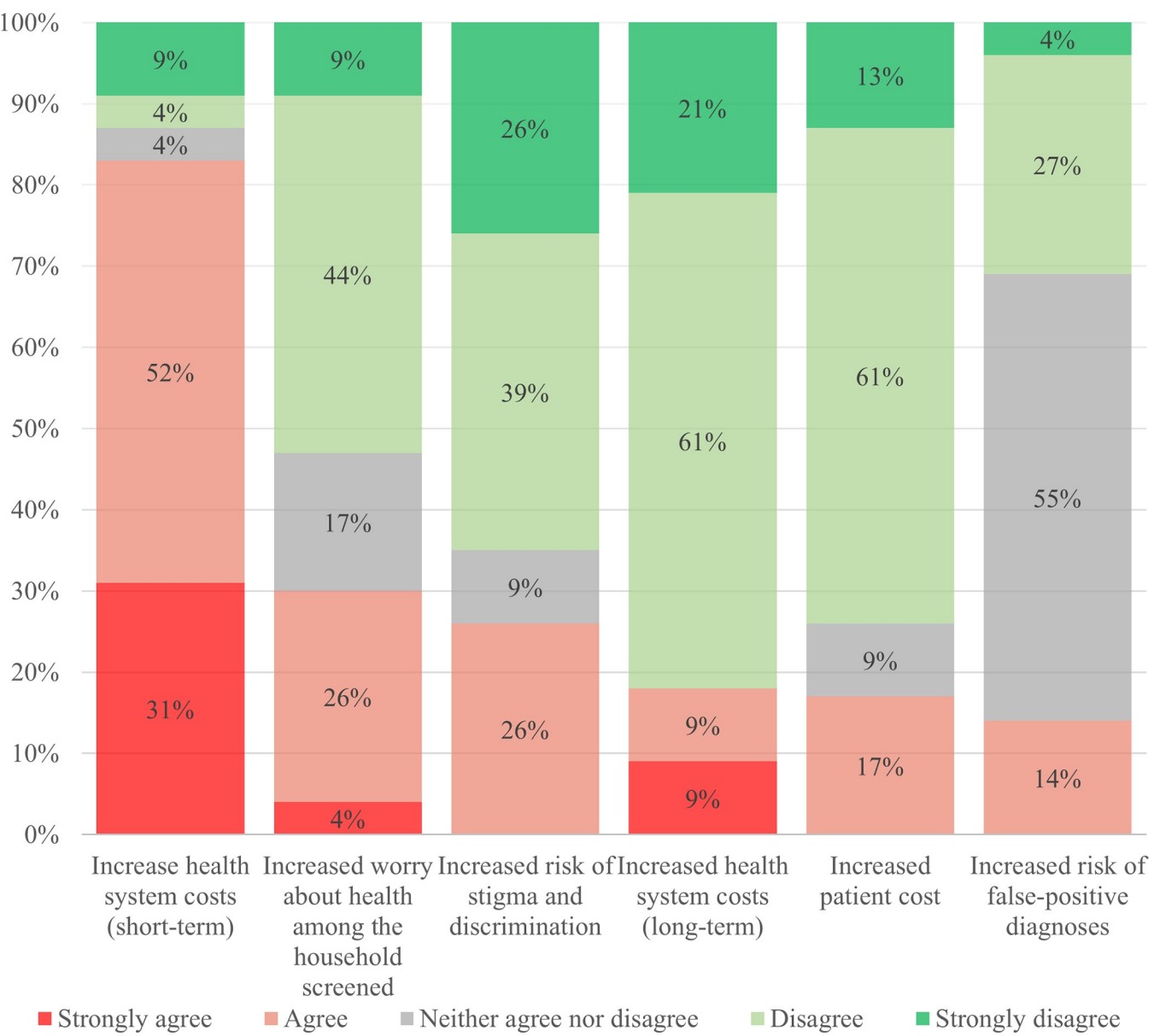

**Fig 4. Perceived risks of active case-finding.**

**Contextual factors influencing ACF policy development and implementation.** A wide variety of factors play a role in both policy development and implementation according to the NTP managers. Some factors are important across many settings, e.g. the country and health system context, and funders' priorities.

Fifty-five percent (n = 12/22) of the NTP managers articulated that factors at the level of the overall country context influenced the development of ACF policies to a high degree. They highlighted the influence of politics, including the political system, commitment, agendas, political change and turnover. In terms of ACF policy implementation, country-level factors also influenced to a high degree according to 45% (n = 10/22) of the NTP managers. Such factors included local geography, climate and culture.

The majority of the NTP managers said that health system factors influenced ACF policy development and implementation to high degrees (77%, n = 17/22; and 68%, n = 15/22, respectively). Most NTP managers emphasized the importance of human and financial resources for ACF policy implementation and buy-in at sub-national level. Focusing on human resources, many NSPs documented the need to not only expand human resources, but to increase their capacity for ACF, and to improve their coordination, engagement and supervision.

Many NTP managers expressed that factors at the level of organizations and the community moderately influenced the development and implementation of ACF policies; (55%, n = 12/22; and 45%, n = 10/22). They stressed the importance of community engagement to develop good, implementable ACF policies.

Ninety-one percent (n = 21/22) and 86% (n = 20/22) of the NTP managers specified that the priorities of funding organizations influenced ACF policy development and implementation, respectively. According to WHO, the NTP budget in five of the included countries does not include any international funding [6]. The NTP managers elaborated that funders demanded seeing impact on case detection and imposed timelines for ACF activities. Overall, funders were described as influential, but not as powerful as the government in policy development.

> "You have to have your policy first and then you prioritize. So, the funders will have a choice. The policy will influence the funders, but not the other way around." (NTP manager 22)

The views of NTP managers on the influence of individual level factors (e.g. knowledge about TB) on ACF policy development were mixed; 18% (n = 4/22) of the respondents considered such factors as highly influential, another 18% saw no influence at all. The NTP managers described the need for TB patients to be at the center of the discussion when developing ACF policies, whilst they would seldom directly influence policy development processes.

> "What we are trying to do is providing them a service they haven't demanded or didn't know they will get, what we need to think about is their expectation, their fear, and take those onboard when we design the policy." (NTP manager 13)

When it comes to ACF policy implementation, all NTP managers declared that the individual level context would influence it to at least some degree. They elaborated on the importance of willingness and trust from the side of individuals participating in ACF, and their knowledge and awareness of TB.

### Processes

**Types of evidence and frequency of their use for ACF policymaking.** The NTP managers were asked about their use of WHO guidelines, international scientific evidence, national scientific evidence, expert knowledge and personal experience. The findings are summarized in Fig 5, showcasing that all different types of evidence are used in making ACF policies, many of them frequently.

All NTP managers used WHO guidelines. They described using them as a reference document, but also underscored the need for contextualization. Conversely, international scientific evidence was utilized less; 21% (n = 5/23) rarely or never used it. The NTP managers described applying international scientific evidence when local data and evidence were limited, to learn from other countries experiences or when in doubt about WHO guidelines.

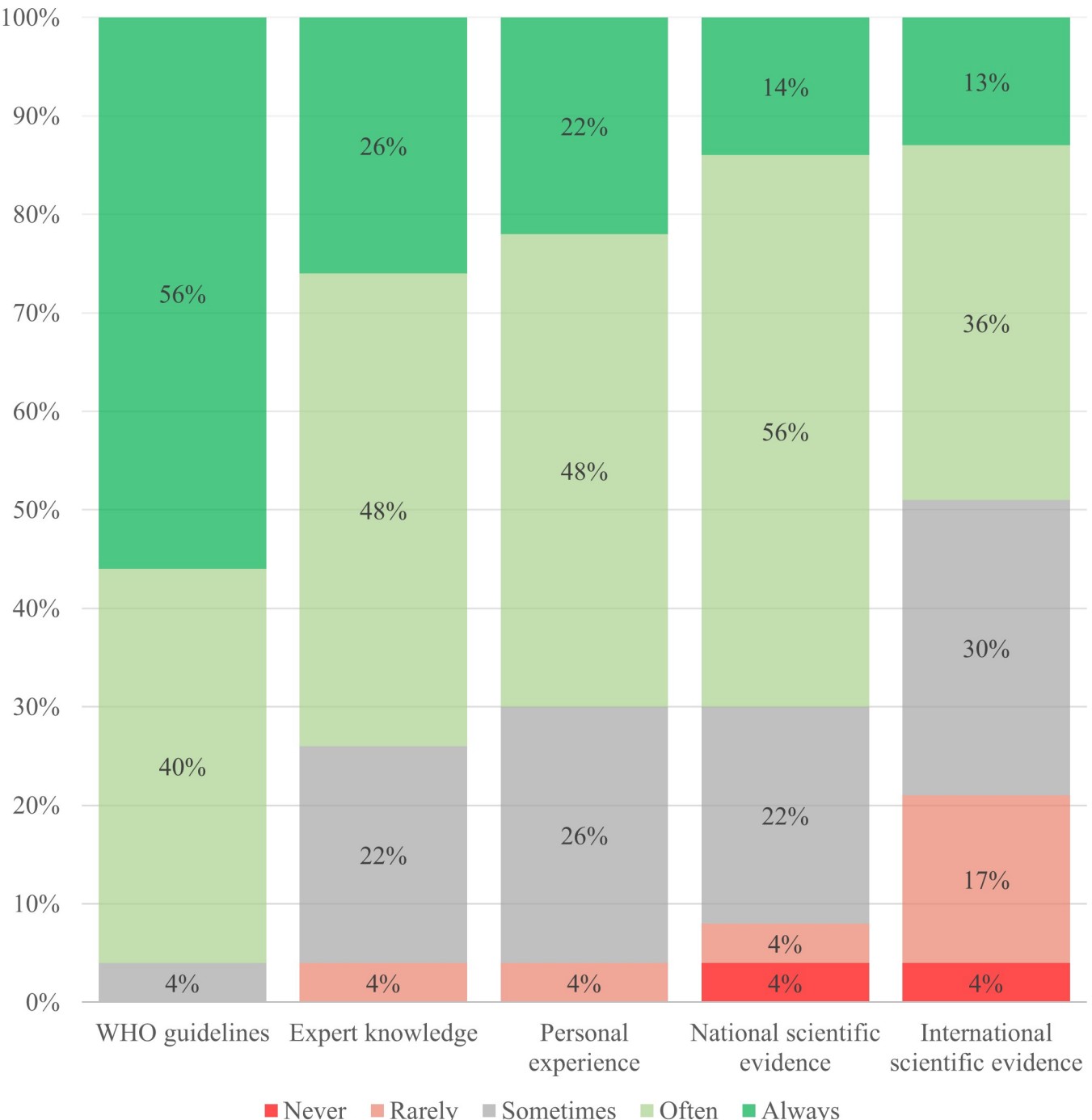

**Fig 5. Frequency of use of evidence in active case-finding policy development and implementation.**

"If we have an argument about what WHO says, then we look for more evidence." (NTP manager 2)

National scientific evidence was also frequently used; 8% (n = 2) of the NTP managers stated they rarely or never utilized it. They stressed the importance of national scientific

evidence (e.g. publications in national journals), especially for implementation. We investigated if upper middle-income countries used national evidence more compared to other countries, given that many have a better national research infrastructure, but replies were mixed. Finally, most NTP managers frequently used expert knowledge and personal experience in ACF policymaking; one NTP manager (4%) asserted not relying on those types of evidence. According to the qualitative data, NTP managers often drew on expert knowledge through institutionalized processes, e.g. regularly convening working groups and committees. Experts could be from a neighboring country with a similar context or from international organizations.

NSPs furthermore described the use of data in NSP development, including from burden of disease analyses, prevalence and drug resistance surveys, and surveys on knowledge, attitudes and practice in ACF policymaking. Besides, the NSPs affirmed the use of country assessments, external program reviews, monitoring reports and lessons learnt for NSP development.

### Actors

**Stakeholders influencing ACF policy development and implementation.**   The 'top three' stakeholders perceived to be influencing ACF policy development were international organizations, policymakers in the national government and managers in a district or region. In terms of ACF policy implementation, the 'top three' stakeholders reported to influence implementation were managers in a healthcare institution (e.g. a hospital), managers in non-governmental organizations and, again, managers in a district or region. Many NTP managers pointed out policymakers in the national government as most powerful stakeholders in ACF policy development and policymakers at sub-national level for implementation.

> "I think the sub-national level, especially the implementers, [are the most powerful] since they have the experience on how to do the activity and we can incorporate the lessons learned based from their experience in the policy to be formulated." (NTP manager 23)

Moreover, the document review underlined implementing partners such as community health workers, volunteers, private providers, pharmacies, religious leaders and setting-specific implementing partners, e.g. prison health workers and wardens.

**Considerations for scaling up ACF.**   All NTP managers articulated that ACF should be scaled up in their countries (100%, n = 22). According to them, scaling up ACF would necessitate political commitment, appropriate diagnostics, data and evidence, community engagement, knowledge and awareness about TB, and human and financial resources. Ninety-one percent (n = 20/22) of the NTP managers proclaimed that financial resources for ACF were insufficient. Nineteen of 20 countries (95%) received support for TB from the Global Fund to Fight AIDS, Tuberculosis and Malaria. When asked about strategies for increasing financial resources, the NTP managers underlined the generation of evidence to show impact, advocacy for domestic funding, diversification of funding sources, and the development and/or exploration of new funding mechanisms. Eighty-nine percent (n = 17/19) of the NTP managers furthermore said that human resources for ACF were inadequate. Strategies for fighting human resource constraints included advocacy for more (and domestically funded) human resources and human resource training.

## Discussion

This study provided insights into ACF policy development, implementation and scale-up in high TB burden countries, based on a mixed-methods survey with NTP managers and a

document review of NSPs. NTP managers considered the national governments the most powerful stakeholder for ACF policy development, while sub-national actors were perceived most powerful for ACF policy implementation. The only 'top' stakeholders named for both ACF policy development and implementation were managers in districts and regions. Different types of evidence were used for developing and implementing ACF policies, while there was a particular demand for local evidence to inform local decisions. Many potential benefits of ACF were highlighted and the need for scale-up was unanimously agreed upon. The NSPs reflected the NTP managers' support of ACF, but not all included explicit aims, targets and target groups related to ACF. The NTP managers acknowledged the risk that ACF may cause increased health system costs in the short-term, but also recognized the possibility of decreasing costs in the long-term. About 90% of the NTP managers declared that financial and human resources were currently lacking, while they also mentioned strategies to overcome resource constraints.

Many of the described factors influencing ACF policy implementation align with the available evidence in the field. Firstly, ACF policy implementation has been widely studied, as documented by a scoping review [17] and more recent studies [24, 25]. In line with our study, a major barrier for ACF implementation is the lack of human and financial resources [26–30]. The lack of resources is also reflected in the World TB Report 2019 [6], which showed that only five of the high TB burden countries had a national TB budget consisting of more than 50% domestic resources. Another five countries had TB budgets largely based on international funding, while the budget of 11 countries remained mostly unfunded [6]. Secondly, the perceived benefits and risks of ACF have been documented in the literature [12, 31–35], while this study provides unique insights into NTP managers' value judgements. The fact that all NTP managers agreed on the need for scaling up ACF seems important given that their leadership and commitment for ACF is considered paramount for ACF policy [11]. The NTP managers commitment to scaling up ACF may be particularly important when TB patients' access to care is impeded, such as during the COVID-19 pandemic.

This study provides new insights regarding key stakeholders and evidence use in ACF policy development and implementation. Firstly, the only 'top' stakeholder named in both ACF policy development and implementation were managers in districts and regions. Their involvement in ACF policy processes may increase ownership and relevance of policies [36], and thus increase the likelihood for implementation. Secondly, besides describing the types of evidence and frequency of evidence use, this study showed that most NTP managers had research experience and many countries seemed to have institutionalized processes for ACF policy development in place, such as regularly convening working groups and committees. Both are indicators conducive to evidence-informed policymaking [37, 38].

NTP managers mentioned strategies to overcome resource constraints for ACF, which appear necessary to implement and scale up ACF. Strategies include a) the diversification of funding sources and b) the generation of local evidence to inform resource allocation for ACF. The importance to invest in local evidence has been highlighted [39], while ACF-related projects offer opportunities for generating context-specific evidence [11]. Meanwhile, local research infrastructure and research capacity may be limited, especially in low- and lower middle-income countries [40, 41]. In terms of diversified funding, our study showed that some countries already have a diverse funding base, including the government, national health insurances, the private sector, etc. Even out-of-pocket payments were explained as a funding source for ACF implementation, while ACF should rather be seen as a strategy to decrease patient cost [42–44]. Demonstrating the effectiveness of ACF investments in relation to other underfunded interventions is important to limit the opportunity cost of scale-up. Adding to the strategies that NTP managers described, ACF could be anchored more strongly in the countries' NSPs, e.g. with concrete targets.

## Future research

Implementation, policy and programmatic research, as well as cost-effectiveness studies on human and financial resource strategies for ACF may be useful to explore the "how to" questions which this survey started to assess. In line with WHO guidance concerning the monitoring and evaluation for ACF, operational research that builds on available local data may help further build the local evidence base to inform local decision-making around ACF. Similar types of surveys and mixed-methods studies targeting sub-national stakeholders may provide relevant local evidence for ACF policy development, implementation and scale-up, complementing the findings of this study.

## Strengths and limitations

For this study, we developed a new questionnaire which allowed us to quantify and qualify the attitudes of NTP managers. We backed up our findings by including information from NSPs. The survey had a high participation rate of 77%. Though innovative, the questionnaire had not been validated and included Likert scales with limited reliability based on Cronbach's Alpha [20]. We aimed to increase the reliability of the NTP managers' responses by implementing the survey as structured interviews. Despite the geographic diversity, we were able to conduct some interviews in person. Those interviews were generally longer, and the quality of the responses may be higher compared to those interviews conducted via phone/Skype. However, the validation process (i.e. sharing the filled-in surveys with the participants) helped us to ensure the quality and depth of the data from all interviews. The survey was conducted in English which was not the native language for many of the participants. This may have led to poorer quality responses. However, English is the working language for most NTP managers, especially in international collaboration that all of them participate in.

In general, survey questionnaires are prone to various types of biases [18]. We attempted to minimize biases by involving two independent researchers from the Evaluation Unit at Karolinska Institutet to critically assess our survey questionnaire. Yet, item social desirability may still have occurred, i.e. in terms of questions being written in such a way as to reflect more socially desirable attitudes [45]. Likewise, mood state bias [45] may be prevalent, i.e. the predisposition of NTP managers to view the topic in certain terms, e.g. generally positive. This may have been influenced by the timing of the survey interview, e.g. NTP managers who participated in the survey while at a WHO meeting might have been in a certain state of mind and away from their day-to-day work. It is possible that questions about the use of evidence may have been affected by this bias, as NTP managers who participated in the survey while at a WHO meeting may have been exposed to WHO guidelines more recently and thus valued them more in their responses. We tried to mitigate these biases by giving NTP managers opportunities to reflect, take breaks and elaborate on their responses.

## Conclusion

The NTP managers' unanimous agreement on the need for ACF scale-up was reflected in the NSPs, but not all NSPs included explicit aims and targets related to ACF. Most NTP managers identified human and financial resource constraints as the main barriers to ACF implementation and scale-up. Strategies to increase resources exist but may not yet have been fully implemented, e.g. generating local evidence for advocacy. Managers in districts and regions were identified as a key stakeholder whose involvement could help improve ACF policy development, implementation and scale-up.

## Supporting information

**S1 Appendix. Document review: Data extraction table.**
(XLSX)

**S2 Appendix. Cross-sectional survey: Questionnaire.**
(DOCX)

**S1 Data.**
(XLSX)

**S2 Data. Dictionary codebook.**
(PDF)

## Acknowledgments

First and foremost, we wish to thank the NTP managers and NTP staff who participated in the survey. We would also like to acknowledge the contributions of Terese Stenfors and Per J Palmgren from Karolinska Institutet in developing the survey questionnaire; of Noemia Teixeira de Siqueira-Filha from the Liverpool School of Tropical Medicine in extracting information in Portuguese language for the document review; of the Stop TB Partnership and WHO for their support in identifying NSPs for the document review; and of WHO for inviting OB to join their meetings to conduct survey interviews.

## Author Contributions

**Conceptualization:** Olivia Biermann, Kerri Viney, Maxine Caws, Knut Lönnroth.

**Data curation:** Olivia Biermann, Phuong Bich Tran, Kristi Sidney Annerstedt.

**Formal analysis:** Olivia Biermann, Phuong Bich Tran, Kristi Sidney Annerstedt.

**Funding acquisition:** Maxine Caws.

**Investigation:** Olivia Biermann, Phuong Bich Tran, Kerri Viney, Knut Lönnroth, Kristi Sidney Annerstedt.

**Methodology:** Olivia Biermann, Kerri Viney, Knut Lönnroth, Kristi Sidney Annerstedt.

**Project administration:** Olivia Biermann.

**Software:** Olivia Biermann.

**Supervision:** Kerri Viney, Maxine Caws, Knut Lönnroth.

**Validation:** Olivia Biermann, Phuong Bich Tran, Kristi Sidney Annerstedt.

**Visualization:** Olivia Biermann.

**Writing – original draft:** Olivia Biermann.

**Writing – review & editing:** Olivia Biermann, Phuong Bich Tran, Kerri Viney, Maxine Caws, Knut Lönnroth, Kristi Sidney Annerstedt.

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
