## [Decision Letter · Decision Letter 0]

23 Jun 2020

PONE-D-20-12519

Active case-finding policy development, implementation and scale-up in high-burden countries: a mixed-methods survey with National Tuberculosis Programme managers and document review

PLOS ONE

Dear Olivia,

Thank you for submitting your manuscript to PLOS ONE. After careful consideration, we feel that it has merit but does not fully meet PLOS ONE’s publication criteria as it currently stands. Therefore, we invite you to submit a revised version of the manuscript that addresses the points raised during the review process.

We look forward to receiving your revised manuscript.

Kind regards,

Hemant Deepak Shewade, MBBS MD

Academic Editor

PLOS ONE

Journal Requirements:

2.We note that you have indicated that data from this study are available upon request. PLOS only allows data to be available upon request if there are legal or ethical restrictions on sharing data publicly. For information on unacceptable data access restrictions, please see http://journals.plos.org/plosone/s/data-availability#loc-unacceptable-data-access-restrictions.

Additional Editor Comments (if provided):

Dear Author (s),

Both the reviewers have provided insightful comments. Please go through the comments carefully and address them.

The length of the manuscript (>5300 words) might reduce the readership of the article. It will be good if the authors can reduce the number of words in the manuscript. There is scope for reducing the words in methods and discussion section of the article.

COVID-19 pandemic associated national lockdowns have disrupted TB services. This has resulted in lockdown associated under-detection of TB. In addition to restoring routine TB services, programmes will have to focus on enhanced and active case finding in high risk populations to detect these people who were missed during lockdown. Please consider discusing the implications of your findings in line with this aspect.

Reviewers' comments:

Reviewer's Responses to Questions

**Comments to the Author**

1. Is the manuscript technically sound, and do the data support the conclusions?

Reviewer #1: Yes

Reviewer #2: Yes

2. Has the statistical analysis been performed appropriately and rigorously? 

Reviewer #1: Yes

Reviewer #2: Yes

3. Have the authors made all data underlying the findings in their manuscript fully available?

Reviewer #1: No

Reviewer #2: Yes

4. Is the manuscript presented in an intelligible fashion and written in standard English?

Reviewer #1: Yes

Reviewer #2: Yes

5. Review Comments to the Author

Reviewer #1: The study explores the attitudes and perception of National TB Programme managers related to the ACF policy development, implementation and scale-up plan provides a good insight on ACF implementation. Also, this manuscript provides a fair idea about the process and actors of policy development and implementation of any newer TB control strategy. Thus, the information from this study could be of help to successfully advocate for adoption of newer strategies like TB preventive therapy for household contacts in high-burden countries.

The authors have managed to comprehensively explore the ACF implementation by adopting mixed-methods design and also including the review of NSPs. Though, I completely acknowledge the vastness and scope of this research work, the manuscript requires is quite lengthy (>5000 words) and requires to be trimmed. There is scope for reducing some words in the manuscript especially in methods and discussion section of the manuscript. Though, in mixed-methods studies, it is difficult to avoid completely the repetition of information between results and discussion, some of it can be deleted from discussion (Line number 479-482)

Below are some specific minor comments.

Abstract

1. In the results section, the component headings like 'Scale-up of ACF' and 'ACF policy development and implementation' can be removed. The results section can be rewritten in the same sequence of ACF policy development, implementation and scale-up, as mentioned in the objectives.

2. The conclusion section can also highlight the need for future operational research to generate local evidence on ACF for policy development.

Introduction:

1. Line 72: You can be explicitly state that about 3 million TB patients were either undetected or undetected but not notified. Also, this estimate of 3 million corresponds to 2018, as the WHO global report of 2019 provides estimates of 2018. Please check.

2. The information on high TB burden countries mentioned in the second paragraph does not give the importance of these countries in line with missing TB patients and also the ACF policy. Thus, it breaks the flow of the introduction. The authors can consider shifting this paragraph just before the aim of the study. Also, can justify the reason for considering only the high TB burden countries.

3. Line 86: I feel the authors can quote here some of the specific research studies which assessed the benefits and challenges of the ACF. I have listed some below.

1. Marks GB, Nguyen NV, Nguyen PTB, et al. Community-wide Screening for Tuberculosis in a High-Prevalence Setting. N Engl J Med. 2019;381(14):1347‐1357. doi:10.1056/NEJMoa1902129

2. Shewade HD, Kumar AMV, Satyanarayana S, et al. Benefits of community-based TB screening vs. passive case finding. Int J Tuberc Lung Dis. 2020;24(4):464‐465. doi:10.5588/ijtld.19.0744

3. Shewade H D, Gupta V, Satyanarayana S, et al. Patient

characteristics, health seeking and delays among new sputum

smear positive TB patients identified through active case finding

when compared to passive case finding in India. PLoS One 2019;

14: e0213345

4. Shewade H D, Gupta V, Satyanarayana S, et al. Active case

finding among marginalised and vulnerable populations reduces

catastrophic costs due to tuberculosis diagnosis. Glob Health

Action 2018; 11: 1494897.

5. Shamanewadi AN, Naik PR, Thekkur P, et al. Enablers and Challenges in the Implementation of Active Case Findings in a Selected District of Karnataka, South India: A Qualitative Study. Tuberc Res Treat. 2020;2020:9746329. Published 2020 Jan 24. doi:10.1155/2020/9746329

6. Dey A, Thekkur P, Ghosh A, et al. Active Case Finding for Tuberculosis through TOUCH Agents in Selected High TB Burden Wards of Kolkata, India: A Mixed Methods Study on Outcomes and Implementation Challenges. Trop Med Infect Dis. 2019;4(4)

Methods:

1. The authors needs to provide reference for the embedded study design as design adopted here doesn't clearly fit into the classical embedded mixed-methods designs. There are two different components- Survey (with quantitative variables and descriptive details for few questions (considered as qualitative)) plus Review of documents. Please provide the reference if the current design is documented in detail elsewhere.

2. Line 110: Were the interviews were always structured? No probes were used to explore more on the issues, when the participants tried to explain their selections (?qualitative data)? Whether this type of interviewing might have limited the exploration on a particular phenomenon?

3. Line 178: Whether second attempt was not made to complete the interview when the interview was discontinued due to bad connection?

4. Line 181: Whether the interviews conducted in English language might have limited the information from the NTP managers of non-native English speaking countries?

5. Line 189: What is the reason for conducting interviews in different settings and mode of communication. Was it based on the convenience of the NTP managers or the investigators? Could this difference in data collection influenced the data obtained?

6. Who conducted the interviews and what was the average duration of these interviews. What proportion of the NTP managers responded to the mail request for validating their responses?

Results:

1. There is lack of consistency in reporting of the numbers and percentages. Sometimes, the percentages are mentioned in words and a few times in numbers. The authors can be consistent in reporting.

2. Line 262: It would be better the mention 'Among the 20 participants who identified themselves either as policymakers or “other”,' instead of 'Among those who identified themselves 262

either as policymakers or “other”,'. The later requires the reader to back calculate the denominator.

3. Line 271: It is important to objectively highlight that the lower percentage agreed for improved treatment outcomes, reduced future health system cost and positive social and economic consequences for TB patient.

4. Line 284: The authors can highlight the lack of agreement or disagreement related to potential

5. Line 361: The heading can be replaced as 'Source of evidence and reference materials for ACF policy-making'

6. Line 401: What does 'national policymakers' mean? Is it policy makers in general health system or the governments? Need clarity.

Discussion:

1. There is quite a bit of repetition from the results. The authors can try to reduce the content in the discussion section.

2. The summary paragraph (first paragraph) can be aligned with the objectives- Policy development, implementation and scale-up.

3. Some of the limitations related to qualitative exploration with structured questionnaire and use of English language for interview could be mentioned.

Reviewer #2: General comments

The authors of this study used a detailed cross-sectional survey administered to 30 high TB burden country’s National Tuberculosis Program (NTP) managers to qualitatively and quantitatively assess policy development, implementation, and scale-up of TB active case finding (ACF). The authors reported that a majority of NTP mangers responded (73%). Survey results highlighted that NTP managers in high burden countries acknowledge the importance of ACF intervention in addressing gaps in TB care, but reported that they face a range of political, resource (including financial), and infrastructural gaps in effectively developing relevant policies, implementing and scaling-up ACF interventions. Evidence from this study can be useful in understanding the current state of NTP decision-making process for ACF and gaps in important policy-relevant evidence that can facilitate effective policy development and implementation of ACF interventions. As such, this reviewer feels that the study provides important information for both the TB researchers and the general public (many of issues that are faced by the NTP managers are akin to the policy development and implementation of many of public health interventions in the resource-poor settings).

Points to address in the revision

While the study is useful in understanding the perspectives of NTP mangers (high-level officials), this in itself is the main limitation of the study. This reviewer acknowledges that the survey design and target respondents are appropriate for the primary focus of this study – conducting a scoping review of high-level factors that influence policy development and implementation/scale-up of active case finding for TB in each country – future studies using similar types of survey method focusing on lower-level officials and local programmatic officers who work much more closely at the field, district, and facility-level may provide important programmatic hurdles in implementing ACF interventions. For example, resource needs, infrastructural challenges faced in different operational settings within a country may vary significantly and these can directly influence how ACF interventions are designed and optimized for their operations, costs, and cost-effectiveness. This reviewer would appreciate it if the authors dedicate a section in the discussion section on this matter (expand a bit in their “Future research” section).

The authors were able to get responses from 23 NTP mangers in this study. Were there certain patterns of responses that may be associated with the policy/health systems/economic indicators of the country of each respondent (see if these have any correlation to the macro-level policy/public health/economic indicator of each country)? The reviewer understands that the study is not powered to investigate such correlations statistically but would be very useful to report on these patterns of responses (e.g. regions, TB burden, NTP budget levels reported to the WHO etc.). For example, NTP mangers in countries where the NTP budget may be highly constrained (and have higher dependency on foreign funding) may have completely different perspective (and therefore, different response to the survey questions) than those with better financial situation. Also, there are countries with very specific ACF (and contact investigation) policies, but due to resource and infrastructural constraints, actual programs may not be adequately operating. Do responses from these NTP mangers (their understanding of existence of these policies and programmatic limitations) match what’s been reported in the public (via policy documents, WHO statements, etc.)?

The authors also state in their limitations (Lines 568 - 575) that participant responses may be affected by the mood state bias. Can the authors provide which of the survey questions may be most affected by this bias and how future research (e.g. if the authors were to conduct this study again, what would they do to reduce this bias?) can improve upon this issue?

Also, it would be important to mention in the study limitations (or anywhere else in the discussion section) that policy/implementation/scale-up issues for ACF may be unique to situations (political, financial, and infrastructural) faced in each country. Therefore, similar types of (improved to include additional questions relevant to the country/setting) survey administered in each country to identify challenges and potential strategies in developing policies, implementing and scaling up ACF interventions would be very important. It would be useful for the authors to mention this as part of their future direction for research.

Based on reporting of the survey results in Figures S3 and 4, this reviewer has one major concern for the survey design. These findings would be been much more useful (again this is my personal opinion and other reviewers may have different perspectives on this matter) if the authors asked the NTP managers to rank each component for the risk factors for ACF and use of evidence in policy making for ACF. For example, it looks clear from Figure S3 that most of the NTP managers feel that increased health systems cost (short-term) is the main perceived risk for ACF intervention, followed by ‘increased worry about health’ and ‘stigma’. Current presentation of survey results in these two figures are a bit difficult to comprehend.

Minor suggestions

Figure S1. Please provide sub-titles describing each of the process a bit more in detail so that readers can have a good understanding just by reviewing the figure and the sub-title itself.

Table 1: It would be good to have a full version of such table presented in the appendix to show the process of developing the survey (this would be very helpful for other researchers to use this study in developing similar types of survey for other disease or for TB within the country)

In Figure S2 (Policy analysis triange), it would be very important to provide sub-text/title for each content of the triangle. For example, it’s not clear from just reviewing the figure itself how these components interact with one another. To this reviewer, a policy triangle is better represented as a process diagram showing how each component interacts with one another and describing the contents of each triangle.

6. PLOS authors have the option to publish the peer review history of their article (what does this mean?). If published, this will include your full peer review and any attached files.

Reviewer #1: No

Reviewer #2: Yes: Hojoon Sohn

---

## [Author Response · Author response to Decision Letter 0]

20 Jul 2020

Point-by-point reviewers’ comments and authors’ answers

Title of the manuscript: “Active case-finding policy development, implementation and scale-up in high-burden countries: a mixed-methods survey with National Tuberculosis Programme managers and document review” 

# Editorial requests and reviewers’ comments Authors’ response

 Editorial requests 

1 Please ensure that your manuscript meets PLOS ONE's style requirements, including those for file naming. See here and here.

We have carefully considered PLOS ONE’s style requirements and ensured our manuscript meets them. 

2 We note that you have indicated that data from this study are available upon request. PLOS only allows data to be available upon request if there are legal or ethical restrictions on sharing data publicly. See here. If there are ethical or legal restrictions on sharing a de-identified data set, please explain them in detail (e.g., data contain potentially identifying or sensitive patient information) and who has imposed them (e.g., an ethics committee). Please also provide contact information for a data access committee, ethics committee, or other institutional body to which data requests may be sent. If there are no restrictions, please upload the minimal anonymized data set necessary to replicate your study findings as either Supporting Information files or to a stable, public repository and provide us with the relevant URLs, DOIs, or accession numbers. Please see here for guidelines on how to de-identify and prepare clinical data for publication. For a list of acceptable repositories, please see here.

We have now included the raw data set without identifying information, answers to open-ended questions, or indicators we did not use in the manuscript. 

We have also clarified the Data Availability Statement: “This is a qualitative study based on a sample of National Tuberculosis Programme managers from high tuberculosis burden countries. The respondents are well-known in the fields of tuberculosis and active case-finding; making the full data set publicly available would breach their privacy. The informed consent that all respondents signed promised full anonymity. The raw data set is included as a Supporting Information file without identifying information, qualitative data or indicators that were not used in the analysis. Following data requests, survey transcripts will be reviewed for any potential identifying information and will only be made available to researchers who sign a data sharing agreement. Data requests may be sent to maike.winters@ki.se.” 

3 The length of the manuscript (>5300 words) might reduce the readership of the article. It will be good if the authors can reduce the number of words in the manuscript. There is scope for reducing the words in methods and discussion section of the article. We have reduced the manuscript by approximately 1,000 words. The word count is 4,357. As suggested, we have mainly worked on shortening the methods and discussion sections. 

4 COVID-19 pandemic associated national lockdowns have disrupted TB services. This has resulted in lockdown associated under-detection of TB. In addition to restoring routine TB services, programmes will have to focus on enhanced and active case finding in high risk populations to detect these people who were missed during lockdown. Please consider discussing the implications of your findings in line with this aspect. We have added this aspect to the Discussion section: “The NTP managers commitment to scaling up ACF may be particularly important when TB patients’ access to care is impeded, such as during the COVID-19 pandemic.” (p. 24, lines 529-530)

Reviewer #1 

5 The manuscript is quite lengthy (>5000 words) and requires to be trimmed. There is scope for reducing some words in the manuscript especially in methods and discussion section of the manuscript. Though, in mixed-methods studies, it is difficult to avoid completely the repetition of information between results and discussion, some of it can be deleted from discussion (Line number 479-482) We have reduced the word count to 4,357 words by mainly shortening the methods and discussion sections, including line numbers 479-482 (as suggested). 

 Abstract 

6 In the results section, the component headings like 'Scale-up of ACF' and 'ACF policy development and implementation' can be removed. The results section can be rewritten in the same sequence of ACF policy development, implementation and scale-up, as mentioned in the objectives. We have rewritten the results section accordingly: “23 of the 30 NTP managers (77%) participated in the survey and 22 (73%) national TB strategic plans were reviewed. NTP managers considered managers in districts and regions key stakeholders for both ACF policy development and implementation. Different types of evidence were used to inform ACF policy, while there was a particular demand for local evidence. The NSPs reflected the NTP managers’ unanimous agreement on the need for ACF scale-up, but not all included explicit aims and targets related to ACF. The NTP managers recognized that ACF may decrease health systems costs in the long-term, while acknowledging the risk for increased health system costs in the short-term. About 90% of the NTP managers declared that financial and human resources were currently lacking, while they also elaborated on strategies to overcome resource constraints.” (p. 2, lines 40-49)

7 The conclusion section can also highlight the need for future operational research to generate local evidence on ACF for policy development. We have now highlighted this aspect as proposed: “Strategies to increase resources exist but may have not yet been fully implemented, e.g. generating local evidence including from operational research for advocacy.” (p. 3, line 64)

 Introduction 

8 Line 72: You can be explicitly state that about 3 million TB patients were either undetected or undetected but not notified. Also, this estimate of 3 million corresponds to 2018, as the WHO global report of 2019 provides estimates of 2018. Please check. We have revised the sentence accordingly: “Due to a combination of underreporting of detected cases and underdiagnosis, there is still a gap of three million between estimated incident TB cases and those notified worldwide [6].” The 3 million indeed corresponds to 2018, not 2019. (p. 5, lines 83-84) 

9 The information on high TB burden countries mentioned in the second paragraph does not give the importance of these countries in line with missing TB patients and also the ACF policy. Thus, it breaks the flow of the introduction. The authors can consider shifting this paragraph just before the aim of the study. Also, can justify the reason for considering only the high TB burden countries. We have shifted and rephrased the information as suggested: “This study focuses on the WHO has defined a list of 30 high TB burden countries, which as they account for 85–89% of the global TB burden [7].” 

10 Line 86: I feel the authors can quote here some of the specific research studies which assessed the benefits and challenges of the ACF. I have listed some below.

1. Marks GB, Nguyen NV, Nguyen PTB, et al. Community-wide Screening for Tuberculosis in a High-Prevalence Setting. N Engl J Med. 2019;381(14):1347‐1357. 

2. Shewade HD, Kumar AMV, Satyanarayana S, et al. Benefits of community-based TB screening vs. passive case finding. Int J Tuberc Lung Dis. 2020;24(4):464‐465. 

3. Shewade H D, Gupta V, Satyanarayana S, et al. Patient characteristics, health seeking and delays among new sputum smear positive TB patients identified through active case finding

when compared to passive case finding in India. PLoS One 2019;14: e0213345

4. Shewade H D, Gupta V, Satyanarayana S, et al. Active case finding among marginalised and vulnerable populations reduces catastrophic costs due to tuberculosis diagnosis. Glob Health

Action 2018; 11: 1494897.

5. Shamanewadi AN, Naik PR, Thekkur P, et al. Enablers and Challenges in the Implementation of Active Case Findings in a Selected District of Karnataka, South India: A Qualitative Study. Tuberc Res Treat. 2020;2020:9746329. Published 2020 Jan 24. 

6. Dey A, Thekkur P, Ghosh A, et al. Active Case Finding for Tuberculosis through TOUCH Agents in Selected High TB Burden Wards of Kolkata, India: A Mixed Methods Study on Outcomes and Implementation Challenges. Trop Med Infect Dis. 2019;4(4). We have included the suggested as well as additional relevant research studies, in the revised Discussion section: “ACF policy implementation has been widely studied, as documented by a scoping review [17] and more recent studies [24,25]. In line with our study, a major barrier for ACF implementation is the lack of human and financial resources [26-30]. The lack of resources is also reflected in the World TB Report 2019 [6], which showed that only five of the high TB burden countries had a national TB budget consisting of more than 50% domestic resources. Another five countries had TB budgets largely based on international funding, while the budget of 11 countries remained mostly unfunded [6]. Secondly, the perceived benefits and risks of ACF have been documented in the literature [12,31-35], while this study provides unique insights into NTP mangers’ value judgements.” (p. 23-24, lines 517-525)

The newly included studies are references 24-35. (p. 34-36, lines 777-814)

The following reference was already included in the original draft (now reference 44): Shewade H D, Gupta V, Satyanarayana S, et al. Active case finding among marginalised and vulnerable populations reduces catastrophic costs due to tuberculosis diagnosis. Glob Health Action 2018; 11: 1494897. (p. 37, lines 838-841)

 Methods 

11 The authors needs to provide reference for the embedded study design as design adopted here doesn't clearly fit into the classical embedded mixed-methods designs. There are two different components- Survey (with quantitative variables and descriptive details for few questions (considered as qualitative)) plus Review of documents. Please provide the reference if the current design is documented in detail elsewhere. We have rephrased and referenced this accordingly: “This was a mixed-methods study with an embedded design (Fig 1) [15], complemented by a document review.” (p. 6, lines 120-121) 

12 Line 110: Were the interviews always structured? No probes were used to explore more on the issues, when the participants tried to explain their selections (?qualitative data)? Whether this type of interviewing might have limited the exploration on a particular phenomenon? We wish to confirm that the interviews were indeed always structured. We have clarified this: “We implemented the all surveys through structured interviews…” (p. 7, line 126)

No probes were used. We have clarified this here: “Question formats included Likert scales, lists, yes/no questions and open-ended questions (without probing questions).” (p. 9, line 167-168). Not using probes may have indeed limited the exploration of the open-ended questions more broadly. While we have highlighted the need for further research to explore some of the topics raised, we have also highlighted in the Limitations section how we think we may have compensated for this limitation: “…the validation process (i.e. sharing the filled-in surveys with the participants) helped us to ensure the quality and depth of the data from all interviews.” (p. 29, lines 666-668)

13 Line 178: Whether second attempt was not made to complete the interview when the interview was discontinued due to bad connection? We have added that “several attempts were made to complete the survey but remained unsuccessful due to limited interest and/or time.” (p. 10, line 200-201) Indeed, we did try to set up follow-up calls via telephone or Skype. We also tried to schedule a physical meeting during an international conference and offered the possibility for the survey to be filled in online. However, none of these attempts were successful, which we suspect was due to limited interest and/or time. 

14 Line 189: What is the reason for conducting interviews in different settings and mode of communication. Was it based on the convenience of the NTP managers or the investigators? Could this difference in data collection influenced the data obtained? We have clarified the reason accordingly: “The primary investigator (OB) conducted Ten ten interviews were conducted in person during international conferences, five at a WHO meeting, and eight on the phone/Skype and five in person at a WHO meeting, depending on the participants’ preferences.” (p. 11, lines 212-214)

Ideally, we would have liked to conduct all interviews in person. Unfortunately, this was not feasible due to limited time and/or resources. We have now highlighted the implications of that as follows: 

“Despite the geographic diversity, we were able to conduct some interviews in person. Those interviews were generally longer, and the quality of the responses may be even higher compared to those interviews conducted via phone/Skype. However, the validation process (i.e. sharing the filled-in surveys with the participants) helped us to ensure the quality and depth of the data from all interviews.” (p. 29, lines 663-668)

15 Who conducted the interviews and what was the average duration of these interviews. What proportion of the NTP managers responded to the mail request for validating their responses? We have clarified these issues as follows: “The primary investigator (OB) conducted…” (p. 11, line 212). The interviews took on average 48 minutes (28-68 minutes), which we now state on p. 11, lines 209-210. We have also added that “Five NTP managers (22%) replied after having received the filled-in survey; one included additional information.” (p. 11, lines 218-219)

 Results 

16 There is lack of consistency in reporting of the numbers and percentages. Sometimes, the percentages are mentioned in words and a few times in numbers. The authors can be consistent in reporting. We have made the revisions as suggested to ensure consistency, mentioning the percentages in numbers. However, with the exception if the number was the first word in the sentence. Then it was presented as a word to be consistent with international scientific reporting standards.

17 Line 262: It would be better the mention 'Among the 20 participants who identified themselves either as policymakers or “other”,' instead of 'Among those who identified themselves 262 either as policymakers or “other”,'. The later requires the reader to back calculate the denominator. We have rephrased this sentence as proposed. (p. 14, lines 284-285)

18 Line 271: It is important to objectively highlight that the lower percentage agreed for improved treatment outcomes, reduced future health system cost and positive social and economic consequences for TB patient. We have reworded this to objectively at several occasions: “Only One NTP manager (4%) disagreed…” (p. 14, lines 293-295) 

“Only Three (14%) (two lower middle-income countries and one low-income country) specified targets for the number of people tested.” (p. 16, line 337)

“Only One NTP manager (4%) asserted not relying on those types of evidence.” (p. 19, lines 420)

19 Line 284: The authors can highlight the lack of agreement or disagreement related to potential risks We have rewritten the sentence to reflect that there was indeed less agreement on the potential risks of ACF: “The NTP managers’ views on potential risks of ACF were more heterogenous compared to their views on the potential benefits (Fig 4).” (p. 15, lines 311-312)

20 Line 361: The heading can be replaced as 'Source of evidence and reference materials for ACF policy-making' We have changed the heading, while trying to stick to the wording used in the survey (“Types of evidence” rather than “Sources of evidence”) and emphasizing that this section is not only about the types of evidence, but also about the frequency of their use: “Types of evidence and frequency of their use of evidence in for ACF policymaking.” (p. 19, line 395-396)

21 Line 401: What does 'national policymakers' mean? Is it policy makers in general health system or the governments? Need clarity. We have clarified that we are referring to “policymakers in the national government”. (p. 20, line 443) 

 Discussion 

22 There is quite a bit of repetition from the results. The authors can try to reduce the content in the discussion section. We have revised the discussion section accordingly by shortening it, by structuring it more clearly and embedding it better in the existing literature (p. 22-29, lines 478-684). 

23 The summary paragraph (first paragraph) can be aligned with the objectives- Policy development, implementation and scale-up. We have revised the paragraph to align with the objectives more clearly: “This study provided insights into the context, processes and actors that play key roles for ACF policy development, and implementation and scale-up in high TB burden countries, based on a mixed-methods survey with NTP managers and a document review of NSPs. NTP managers considered the national governments the most powerful stakeholder for ACF policy development, while sub-national actors were perceived most powerful for ACF policy implementation. The only ‘top’ stakeholder named for both ACF policy development and implementation were managers in districts and regions. Different types of evidence were used for developing and implementing ACF policies, while there was a particular demand for local evidence to inform local decisions. Many potential benefits of ACF were highlighted and the need for scale-up was unanimously agreed upon. The NSPs reflected the NTP managers’ support of ACF, but not all included explicit aims, targets and target groups related to ACF. The NTP managers acknowledged the risk that ACF may cause increased health system costs in the short-term, but also recognized the possibility of decreasing costs in the long-term. About 90% of the NTP managers declared that financial and human resources were currently lacking, while they also mentioned strategies to overcome resource constraints.” (p. 22, lines 480-494) 

24 Some of the limitations related to qualitative exploration with structured questionnaire and use of English language for interview could be mentioned. We have elaborated on these limitations as suggested: “Despite the geographic diversity, we were able to conduct some interviews in person. Those interviews were generally longer, and the quality of the responses may be even higher compared to those interviews conducted via phone/Skype. However, the validation process (i.e. sharing the filled-in surveys with the participants) helped us to ensure the quality and depth of the data from all interviews. The survey was conducted in English which was not the native language for many of the participants. This may have led to poorer quality responses. However, English is the working language for most NTP managers.” (p. 29-30, lines 663-670)

Reviewer #2

25 Future studies using similar types of survey method focusing on lower-level officials and local programmatic officers who work much more closely at the field, district, and facility-level may provide important programmatic hurdles in implementing ACF interventions. For example, resource needs, infrastructural challenges faced in different operational settings within a country may vary significantly and these can directly influence how ACF interventions are designed and optimized for their operations, costs, and cost-effectiveness. This reviewer would appreciate it if the authors dedicate a section in the discussion section on this matter (expand a bit in their “Future research” section). We have expanded on the Future research section as suggested, referring broadly to “sub-national stakeholders”, who we agree would be very important to include: “Similar types of surveys and mixed-methods studies targeting sub-national stakeholders may provide relevant local evidence for ACF policy development, implementation and scale-up, complementing the findings of this study.” (p. 29, lines 649-652)

26 The authors were able to get responses from 23 NTP mangers in this study. Were there certain patterns of responses that may be associated with the policy/health systems/economic indicators of the country of each respondent (see if these have any correlation to the macro-level policy/public health/economic indicator of each country)? The reviewer understands that the study is not powered to investigate such correlations statistically but would be very useful to report on these patterns of responses (e.g. regions, TB burden, NTP budget levels reported to the WHO etc.). For example, NTP mangers in countries where the NTP budget may be highly constrained (and have higher dependency on foreign funding) may have completely different perspective (and therefore, different response to the survey questions) than those with better financial situation. We have expanded our analysis to investigate such patterns as suggested. Taking indicators from the World Bank and the World Health Organizations, we tried to give more nuance to the results. First: “We added additional indicators to investigate any patterns in the responses: 1) country income level and 2) region [21], and 3) proportion of NTP budget consisting of domestic funding, 4) international funding and 5) being unfunded [6].” (p. 12, lines 232-234)

We were not able to find any patterns, but have highlighted those new indicators where appropriate in the Results and Discussion sections, e.g. “One NTP manager (4%) from a lower middle-income country disagreed that ACF leads to improved treatment outcomes, while another NTP manager (4%) from a low-income country disagreed that ACF leads to reduced future health system cost, and positive social and economic consequences for a TB patient.” (p. 14-15, lines 295-297). 

“Four of the five upper middle-income countries (80%) included in this study were among those who neither agreed nor disagreed.” (p. 15, lines 315-316)

“…both estimates included NTP managers from low-, lower middle- and upper middle-income countries.” (p. 15, lines 319-320) 

“Three (14%) (two lower middle-income countries and one low-income country) specified targets for the number of people tested.” (p. 16, lines 227-338)

“According to WHO, the NTP budget in five of the included countries does not include any international funding [6].” (p. 17, lines 369-370)

“We investigated if upper middle-income countries used national evidence more compared to other countries, given that many have a better national research infrastructure, but replies were mixed.” (p. 19, lines 416-419)

“The lack of resources is also reflected in the World TB Report 2019 [6], which showed that only five of the high TB burden countries had a national TB budget consisting of more than 50% domestic resources. Another five countries had TB budgets largely based on international funding, while the budget of 11 countries remained mostly unfunded [6].” (p. 24, lines 520-524)

27 There are countries with very specific ACF (and contact investigation) policies, but due to resource and infrastructural constraints, actual programs may not be adequately operating. Do responses from these NTP mangers (their understanding of existence of these policies and programmatic limitations) match what’s been reported in the public (via policy documents, WHO statements, etc.)? The responses from NTP managers with regards resource constraints do match what has been reported in some National TB Strategic Plans, e.g. p. 17, lines 357-359: “Focusing on human resources, many NSPs documented the need to not only expand human resources, but to increase their capacity for ACF, and to improve their coordination, engagement and supervision.”. 

Moreover, while we have reported the NTP managers qualitative answers on resource constraints in the Results section, Appendix 1 “Data extraction table” provides further insights into how the NSPs have reflected operational considerations, e.g. health system requirements and other operational consideration (columns Q-T). 

We agree that it would be very interesting and valuable to explore whether NTP managers’ responses match what has been reported in public (beyond the NSPs) and hope that future studies may address this important issue.

28 The authors also state in their limitations (Lines 568 - 575) that participant responses may be affected by the mood state bias. Can the authors provide which of the survey questions may be most affected by this bias and how future research (e.g. if the authors were to conduct this study again, what would they do to reduce this bias?) can improve upon this issue? We have added some information on the questions which we think may have been more affected by mood state bias than others: “It is possible that questions about the use of evidence may have been affected by this bias, as NTP managers who participated in the survey while at a WHO meeting may have been exposed to WHO guidelines more recently and thus valued them more in their responses.” (p. 30, lines 681-684)

In cases where mood state bias may have been caused by the location of the interview, future research may improve upon this issue by conducting interviews under similar conditions, as much as possible. Meanwhile, we acknowledge that making participation convenient for respondents required us to conduct the interviews in different settings and that the participation rate may have otherwise dropped considerably. In the case of NTP managers or other high-level officials, we believe it may be better to focus on convenience of participation and accept risk of mood state bias. 

29 Also, it would be important to mention in the study limitations (or anywhere else in the discussion section) that policy/implementation/ scale-up issues for ACF may be unique to situations (political, financial, and infrastructural) faced in each country. Therefore, similar types of (improved to include additional questions relevant to the country/setting) survey administered in each country to identify challenges and potential strategies in developing policies, implementing and scaling up ACF interventions would be very important. It would be useful for the authors to mention this as part of their future direction for research. We have reflected this in the Future research section, as we fully agree that developing a better contextual understanding for ACF policy development, implementation and scale-up will be of utmost importance moving forward: “Similar types of surveys and mixed-methods studies targeting sub-national stakeholders may provide relevant local evidence for ACF policy development, implementation and scale-up, complementing the findings of this study.” (p. 29, lines 649-652)

30 Based on reporting of the survey results in Figures S3 and 4, this reviewer has one major concern for the survey design. These findings would be been much more useful (again this is my personal opinion and other reviewers may have different perspectives on this matter) if the authors asked the NTP managers to rank each component for the risk factors for ACF and use of evidence in policy making for ACF. For example, it looks clear from Figure S3 that most of the NTP managers feel that increased health systems cost (short-term) is the main perceived risk for ACF intervention, followed by ‘increased worry about health’ and ‘stigma’. Current presentation of survey results in these two figures are a bit difficult to comprehend. To make Figures 3 and 4 more comprehensive, we have changed the order of the bars aiming to show which aspects NTP managers agreed with most to least. We furthermore agree that future surveys could change the design of these questions to rankings rather than ratings. However, it may be important to note that based on our experience of having included one ranking question (2.4 “How would you rank the importance of ACF among other TB interventions for early case detection? Please rank from 1-7 [1 = most important]), we felt that this type of question seemed considerably more difficult and less intuitive for respondents to answer. In fact, many respondents chose to skip this questions as it took too long consider. This is why data on this question remained incomplete and we did not include it in the final analysis. 

31 Figure S1. Please provide sub-titles describing each of the process a bit more in detail so that readers can have a good understanding just by reviewing the figure and the sub-title itself. We have provided sub-titles as suggested. For Figure 1: “23 National Tuberculosis (TB) Programme managers participated in the cross-sectional survey, which was analyzed using descriptive statistics and manifest content analysis. 22 National TB Strategic Plans were reviewed, and data extracted to complement the survey findings. All data were analyzed in parallel and merged in the interpretation of the findings.” (p. 7, text box, line 30) 

32 Table 1: It would be good to have a full version of such table presented in the appendix to show the process of developing the survey (this would be very helpful for other researchers to use this study in developing similar types of survey for other disease or for TB within the country) We think that the survey questionnaire (Appendix 2) may be most helpful for other researchers to use for developing similar types of surveys for other diseases or for TB within the country.

As the full qualitative analysis table contains meaning units which are direct quotes from the open-ended questions and may compromise NTP managers anonymity, we are, unfortunately, unable to share the table as ethical restrictions apply. 

33

 In Figure S2 (Policy analysis triangle), it would be very important to provide sub-text/title for each content of the triangle. For example, it’s not clear from just reviewing the figure itself how these components interact with one another. To this reviewer, a policy triangle is better represented as a process diagram showing how each component interacts with one another and describing the contents of each triangle. We have provided sub-titles for Figure 2: “This study showed that the content of active case-finding (ACF) policies, including their development, implementation and scale-up, is influenced by context, actors and processes. Example topics: Context – Important factors included the perceptions of the benefits and risks of ACF and the availability of resources. Actors – Key actors comprised the national government, managers in districts and regions and non-governmental organizations (NGOs). Process – Important processes were evidence use and institutionalized processes to facilitate the same, e.g. regularly convening working groups.” (p. 8, text box, line 144)

We have constructed Figure 2 to give an insight into factors that play a role in ACF policy development, implementation and scale-up across the high TB burden countries, highlighting a few examples. Yet, we strongly agree that the interaction between the different factors that influence ACF policy development, implementation and scale-up would be important to investigate. Such an analysis may provide an in-depth understanding of these processes in a given context and we hope future studies may address this.

---

## [Editor Report · Decision Letter 1]

5 Aug 2020

PONE-D-20-12519R1

Active case-finding policy development, implementation and scale-up in high-burden countries: a mixed-methods survey with National Tuberculosis Programme managers and document review

PLOS ONE

Dear Olivia,

Kindly resubmit.

The response to review comments is not formatted well enough to differentiate between reviewer comment and author resposne.

I suggest this

Mention in capital REVIEWER COMMENT before a reviewer comment and then follow it up with AUTHOR RESPONSE in capital before providing a response.

Give a line space between one set of REVIEWER COMMENT and AUTHOR RESPONSE.

Do this uniformly throughout the response to reviewer comment file.

In the author response, please mention the exact line where the edits have been made in the revised mansucript with track changes.

Please resubmit by 10 Aug 2020.

Kind regards,

Hemant Deepak Shewade, MBBS MD

Academic Editor

PLOS ONE

---

## [Author Response · Author response to Decision Letter 1]

7 Aug 2020

EDITORIAL REQUEST

1. Please ensure that your manuscript meets PLOS ONE's style requirements, including those for file naming. See here and here.

AUTHOR RESPONSE

We have carefully considered PLOS ONE’s style requirements and ensured our manuscript meets them. 

EDITORIAL REQUEST

2. We note that you have indicated that data from this study are available upon request. PLOS only allows data to be available upon request if there are legal or ethical restrictions on sharing data publicly. See here. If there are ethical or legal restrictions on sharing a de-identified data set, please explain them in detail (e.g., data contain potentially identifying or sensitive patient information) and who has imposed them (e.g., an ethics committee). Please also provide contact information for a data access committee, ethics committee, or other institutional body to which data requests may be sent. If there are no restrictions, please upload the minimal anonymized data set necessary to replicate your study findings as either Supporting Information files or to a stable, public repository and provide us with the relevant URLs, DOIs, or accession numbers. Please see here for guidelines on how to de-identify and prepare clinical data for publication. For a list of acceptable repositories, please see here.

AUTHOR RESPONSE

We have now included the raw data set without identifying information, answers to open-ended questions, or indicators we did not use in the manuscript. 

We have also clarified the Data Availability Statement: “This is a mixed-methods study based on a sample of National Tuberculosis Programme managers from high tuberculosis burden countries. The respondents are well-known in the fields of tuberculosis and active case-finding; making the full data set publicly available would breach their privacy. The informed consent that all respondents signed promised full anonymity. The raw data set is included as a Supporting Information file without identifying information, qualitative data or indicators that were not used in the analysis. Following data requests, survey transcripts will be reviewed for any potential identifying information and will only be made available to researchers who sign a data sharing agreement. Data requests may be sent to maike.winters@ki.se.”

EDITORIAL REQUEST

3. The length of the manuscript (>5300 words) might reduce the readership of the article. It will be good if the authors can reduce the number of words in the manuscript. There is scope for reducing the words in methods and discussion section of the article. 

AUTHOR RESPONSE

We have reduced the manuscript by approximately 1,000 words. The word count is 4,357. As suggested, we have mainly worked on shortening the methods and discussion sections. 

EDITORIAL REQUEST

4. COVID-19 pandemic associated national lockdowns have disrupted TB services. This has resulted in lockdown associated under-detection of TB. In addition to restoring routine TB services, programmes will have to focus on enhanced and active case finding in high risk populations to detect these people who were missed during lockdown. Please consider discussing the implications of your findings in line with this aspect. 

AUTHOR RESPONSE

We have added this aspect to the Discussion section: “The NTP managers commitment to scaling up ACF may be particularly important when TB patients’ access to care is impeded, such as during the COVID-19 pandemic.” (p. 24, lines 529-530)

REVIEWER 1 COMMENT

5. The manuscript is quite lengthy (>5000 words) and requires to be trimmed. There is scope for reducing some words in the manuscript especially in methods and discussion section of the manuscript. Though, in mixed-methods studies, it is difficult to avoid completely the repetition of information between results and discussion, some of it can be deleted from discussion (lines 479-482) 

AUTHOR RESPONSE

We have reduced the word count to 4,357 words by mainly shortening the methods and discussion sections, including line numbers 479-482 (as suggested). 

REVIEWER 1 COMMENT

6. In the results section, the component headings like 'Scale-up of ACF' and 'ACF policy development and implementation' can be removed. The results section can be rewritten in the same sequence of ACF policy development, implementation and scale-up, as mentioned in the objectives. 

AUTHOR RESPONSE 

We have rewritten the results section accordingly: “23 of the 30 NTP managers (77%) participated in the survey and 22 (73%) national TB strategic plans were reviewed. NTP managers considered managers in districts and regions key stakeholders for both ACF policy development and implementation. Different types of evidence were used to inform ACF policy, while there was a particular demand for local evidence. The NSPs reflected the NTP managers’ unanimous agreement on the need for ACF scale-up, but not all included explicit aims and targets related to ACF. The NTP managers recognized that ACF may decrease health systems costs in the long-term, while acknowledging the risk for increased health system costs in the short-term. About 90% of the NTP managers declared that financial and human resources were currently lacking, while they also elaborated on strategies to overcome resource constraints.” (p. 2, lines 40-49)

REVIEWER 1 COMMENT

7. The conclusion section can also highlight the need for future operational research to generate local evidence on ACF for policy development. 

AUTHOR RESPONSE

We have now highlighted this aspect as proposed: “Strategies to increase resources exist but may have not yet been fully implemented, e.g. generating local evidence including from operational research for advocacy.” (p. 3, line 64)

REVIEWER 1 COMMENT

8. Line 72: You can be explicitly state that about 3 million TB patients were either undetected or undetected but not notified. Also, this estimate of 3 million corresponds to 2018, as the WHO global report of 2019 provides estimates of 2018. Please check. 

AUTHOR RESPONSE 

We have revised the sentence accordingly: “Due to a combination of underreporting of detected cases and underdiagnosis, there is still a gap of three million between estimated incident TB cases and those notified worldwide [6].” The 3 million indeed corresponds to 2018, not 2019. (p. 5, lines 83-84) 

REVIEWER 1 COMMENT

9. The information on high TB burden countries mentioned in the second paragraph does not give the importance of these countries in line with missing TB patients and also the ACF policy. Thus, it breaks the flow of the introduction. The authors can consider shifting this paragraph just before the aim of the study. Also, can justify the reason for considering only the high TB burden countries. 

AUTHOR RESPONSE

We have shifted and rephrased the information as suggested: “This study focuses on the 30 high TB burden countries, as they account for 85–89% of the global TB burden [7].” 

REVIEWER 1 COMMENT

10. Line 86: I feel the authors can quote here some of the specific research studies which assessed the benefits and challenges of the ACF. I have listed some below.

1. Marks GB, Nguyen NV, Nguyen PTB, et al. Community-wide Screening for Tuberculosis in a High-Prevalence Setting. N Engl J Med. 2019;381(14):1347‐1357. 

2. Shewade HD, Kumar AMV, Satyanarayana S, et al. Benefits of community-based TB screening vs. passive case finding. Int J Tuberc Lung Dis. 2020;24(4):464‐465. 

3. Shewade H D, Gupta V, Satyanarayana S, et al. Patient characteristics, health seeking and delays among new sputum smear positive TB patients identified through active case finding

when compared to passive case finding in India. PLoS One 2019;14: e0213345

4. Shewade H D, Gupta V, Satyanarayana S, et al. Active case finding among marginalised and vulnerable populations reduces catastrophic costs due to tuberculosis diagnosis. Glob Health

Action 2018; 11: 1494897.

5. Shamanewadi AN, Naik PR, Thekkur P, et al. Enablers and Challenges in the Implementation of Active Case Findings in a Selected District of Karnataka, South India: A Qualitative Study. Tuberc Res Treat. 2020;2020:9746329. Published 2020 Jan 24. 

6. Dey A, Thekkur P, Ghosh A, et al. Active Case Finding for Tuberculosis through TOUCH Agents in Selected High TB Burden Wards of Kolkata, India: A Mixed Methods Study on Outcomes and Implementation Challenges. Trop Med Infect Dis. 2019;4(4). 

AUTHOR RESPONSE

We have included the suggested as well as additional relevant research studies, in the revised Discussion section: “ACF policy implementation has been widely studied, as documented by a scoping review [17] and more recent studies [24,25]. In line with our study, a major barrier for ACF implementation is the lack of human and financial resources [26-30]. The lack of resources is also reflected in the World TB Report 2019 [6], which showed that only five of the high TB burden countries had a national TB budget consisting of more than 50% domestic resources. Another five countries had TB budgets largely based on international funding, while the budget of 11 countries remained mostly unfunded [6]. Secondly, the perceived benefits and risks of ACF have been documented in the literature [12,31-35], while this study provides unique insights into NTP mangers’ value judgements.” (p. 23-24, lines 517-525)

The newly included studies are references 24-35. (p. 34-36, lines 777-814)

The following reference was already included in the original draft (now reference 44): Shewade H D, Gupta V, Satyanarayana S, et al. Active case finding among marginalised and vulnerable populations reduces catastrophic costs due to tuberculosis diagnosis. Glob Health Action 2018; 11: 1494897. (p. 37, lines 838-841)

REVIEWER 1 COMMENT

11. The authors needs to provide reference for the embedded study design as design adopted here doesn't clearly fit into the classical embedded mixed-methods designs. There are two different components- Survey (with quantitative variables and descriptive details for few questions (considered as qualitative)) plus Review of documents. Please provide the reference if the current design is documented in detail elsewhere. 

AUTHOR RESPONSE

We have rephrased and referenced this accordingly: “This was a mixed-methods study with an embedded design (Fig 1) [15], complemented by a document review.” (p. 6, lines 120-121) 

REVIEWER 1 COMMENT

12. Line 110: Were the interviews always structured? No probes were used to explore more on the issues, when the participants tried to explain their selections (?qualitative data)? Whether this type of interviewing might have limited the exploration on a particular phenomenon?

AUTHOR RESPONSE

We wish to confirm that the interviews were indeed always structured. We have clarified this: “We implemented all surveys through structured interviews…” (p. 7, line 126)

No probes were used. We have clarified this here: “Question formats included Likert scales, lists, yes/no questions and open-ended questions (without probing questions).” (p. 9, line 167-168). Not using probes may have indeed limited the exploration of the open-ended questions more broadly. While we have highlighted the need for further research to explore some of the topics raised, we have also highlighted in the Limitations section how we think we may have compensated for this limitation: “…the validation process (i.e. sharing the filled-in surveys with the participants) helped us to ensure the quality and depth of the data from all interviews.” (p. 29, lines 666-668)

REVIEWER 1 COMMENT

13. Line 178: Whether second attempt was not made to complete the interview when the interview was discontinued due to bad connection? 

AUTHOR RESPONSE 

We have added that “several attempts were made to complete the survey but remained unsuccessful due to limited interest and/or time.” (p. 10, line 200-201) Indeed, we did try to set up follow-up calls via telephone or Skype. We also tried to schedule a physical meeting during an international conference and offered the possibility for the survey to be filled in online. However, none of these attempts were successful, which we suspect was due to limited interest and/or time. 

REVIEWER 1 COMMENT

14. Line 189: What is the reason for conducting interviews in different settings and mode of communication. Was it based on the convenience of the NTP managers or the investigators? Could this difference in data collection influenced the data obtained? 

AUTHOR RESPONSE

We have clarified the reason accordingly: “The primary investigator (OB) conducted ten interviews in person during international conferences, five at a WHO meeting, and eight on the phone/Skype, depending on the participants’ preferences.” (p. 11, lines 212-214)

Ideally, we would have liked to conduct all interviews in person. Unfortunately, this was not feasible due to limited time and/or resources. We have now highlighted the implications of that as follows: 

“Despite the geographic diversity, we were able to conduct some interviews in person. Those interviews were generally longer, and the quality of the responses may be even higher compared to those interviews conducted via phone/Skype. However, the validation process (i.e. sharing the filled-in surveys with the participants) helped us to ensure the quality and depth of the data from all interviews.” (p. 29, lines 663-668)

REVIEWER 1 COMMENT

15. Who conducted the interviews and what was the average duration of these interviews. What proportion of the NTP managers responded to the mail request for validating their responses? 

AUTHOR RESPONSE

We have clarified these issues as follows: “The primary investigator (OB) conducted…” (p. 11, line 212). The interviews took on average 48 minutes (28-68 minutes), which we now state on p. 11, lines 209-210. We have also added that “Five NTP managers (22%) replied after having received the filled-in survey; one included additional information.” (p. 11, lines 218-219)

REVIEWER 1 COMMENT

16. There is lack of consistency in reporting of the numbers and percentages. Sometimes, the percentages are mentioned in words and a few times in numbers. The authors can be consistent in reporting. 

AUTHOR RESPONSE

We have made the revisions as suggested to ensure consistency, mentioning the percentages in numbers. However, with the exception if the number was the first word in the sentence. Then it was presented as a word to be consistent with international scientific reporting standards.

REVIEWER 1 COMMENT

17. Line 262: It would be better the mention 'Among the 20 participants who identified themselves either as policymakers or “other”,' instead of 'Among those who identified themselves 262 either as policymakers or “other”,'. The later requires the reader to back calculate the denominator. 

AUTHOR RESPONSE

We have rephrased this sentence as proposed. (p. 14, lines 284-285)

REVIEWER 1 COMMENT

18. Line 271: It is important to objectively highlight that the lower percentage agreed for improved treatment outcomes, reduced future health system cost and positive social and economic consequences for TB patient. 

AUTHOR RESPONSE

We have reworded this to objectively on several occasions: “One NTP manager (4%) disagreed…” (p. 14, lines 293-295) 

“Three (14%) (two lower middle-income countries and one low-income country) specified targets for the number of people tested.” (p. 16, line 337)

“One NTP manager (4%) asserted not relying on those types of evidence.” (p. 19, lines 420)

REVIEWER 1 COMMENT

19. Line 284: The authors can highlight the lack of agreement or disagreement related to potential risks 

AUTHOR RESPONSE

We have rewritten the sentence to reflect that there was indeed less agreement on the potential risks of ACF: “The NTP managers’ views on potential risks of ACF were more heterogenous compared to their views on the potential benefits (Fig 4).” (p. 15, lines 311-312)

REVIEWER 1 COMMENT

20. Line 361: The heading can be replaced as 'Source of evidence and reference materials for ACF policy-making' 

AUTHOR RESPONSE

We have changed the heading, while trying to stick to the wording used in the survey (“Types of evidence” rather than “Sources of evidence”) and emphasizing that this section is not only about the types of evidence, but also about the frequency of their use: “Types of evidence and frequency of their use for ACF policymaking.” (p. 19, line 395-396)

REVIEWER 1 COMMENT

21. Line 401: What does 'national policymakers' mean? Is it policy makers in general health system or the governments? Need clarity. 

AUTHOR RESPONSE

We have clarified that we are referring to “policymakers in the national government”. (p. 20, line 443) 

REVIEWER 1 COMMENT 

22. There is quite a bit of repetition from the results. The authors can try to reduce the content in the discussion section. 

AUTHOR RESPONSE

We have revised the discussion section accordingly by shortening it, by structuring it more clearly and embedding it better in the existing literature (p. 22-29, lines 478-684). 

REVIEWER 1 COMMENT

23. The summary paragraph (first paragraph) can be aligned with the objectives- Policy development, implementation and scale-up. 

AUTHOR RESPONSE

We have revised the paragraph to align with the objectives more clearly: “This study provided insights into the context, processes and actors that play key roles for ACF policy development, and implementation and scale-up in high TB burden countries, based on a mixed-methods survey with NTP managers and a document review of NSPs. NTP managers considered the national governments the most powerful stakeholder for ACF policy development, while sub-national actors were perceived most powerful for ACF policy implementation. The only ‘top’ stakeholder named for both ACF policy development and implementation were managers in districts and regions. Different types of evidence were used for developing and implementing ACF policies, while there was a particular demand for local evidence to inform local decisions. Many potential benefits of ACF were highlighted and the need for scale-up was unanimously agreed upon. The NSPs reflected the NTP managers’ support of ACF, but not all included explicit aims, targets and target groups related to ACF. The NTP managers acknowledged the risk that ACF may cause increased health system costs in the short-term, but also recognized the possibility of decreasing costs in the long-term. About 90% of the NTP managers declared that financial and human resources were currently lacking, while they also mentioned strategies to overcome resource constraints.” (p. 22, lines 480-494) 

REVIEWER 1 COMMENT

24. Some of the limitations related to qualitative exploration with structured questionnaire and use of English language for interview could be mentioned. 

AUTHOR RESPONSE

We have elaborated on these limitations as suggested: “Despite the geographic diversity, we were able to conduct some interviews in person. Those interviews were generally longer, and the quality of the responses may be even higher compared to those interviews conducted via phone/Skype. However, the validation process (i.e. sharing the filled-in surveys with the participants) helped us to ensure the quality and depth of the data from all interviews. The survey was conducted in English which was not the native language for many of the participants. This may have led to poorer quality responses. However, English is the working language for most NTP managers.” (p. 29-30, lines 663-670)

REVIEWER 2 COMMENT

25. Future studies using similar types of survey method focusing on lower-level officials and local programmatic officers who work much more closely at the field, district, and facility-level may provide important programmatic hurdles in implementing ACF interventions. For example, resource needs, infrastructural challenges faced in different operational settings within a country may vary significantly and these can directly influence how ACF interventions are designed and optimized for their operations, costs, and cost-effectiveness. This reviewer would appreciate it if the authors dedicate a section in the discussion section on this matter (expand a bit in their “Future research” section). 

AUTHOR RESPONSE

We have expanded on the Future research section as suggested, referring broadly to “sub-national stakeholders”, who we agree would be very important to include: “Similar types of surveys and mixed-methods studies targeting sub-national stakeholders may provide relevant local evidence for ACF policy development, implementation and scale-up, complementing the findings of this study.” (p. 29, lines 649-652)

REVIEWER 2 COMMENT

26. The authors were able to get responses from 23 NTP mangers in this study. Were there certain patterns of responses that may be associated with the policy/health systems/economic indicators of the country of each respondent (see if these have any correlation to the macro-level policy/public health/economic indicator of each country)? The reviewer understands that the study is not powered to investigate such correlations statistically but would be very useful to report on these patterns of responses (e.g. regions, TB burden, NTP budget levels reported to the WHO etc.). For example, NTP mangers in countries where the NTP budget may be highly constrained (and have higher dependency on foreign funding) may have completely different perspective (and therefore, different response to the survey questions) than those with better financial situation. 

AUTHOR RESPONSE

We have expanded our analysis to investigate such patterns as suggested. Taking indicators from the World Bank and the World Health Organizations, we tried to give more nuance to the results. First: “We added additional indicators to investigate any patterns in the responses: 1) country income level and 2) region [21], and 3) proportion of NTP budget consisting of domestic funding, 4) international funding and 5) being unfunded [6].” (p. 12, lines 232-234)

We were not able to find any patterns, but have highlighted those new indicators where appropriate in the Results and Discussion sections, e.g. “One NTP manager (4%) from a lower middle-income country disagreed that ACF leads to improved treatment outcomes, while another NTP manager (4%) from a low-income country disagreed that ACF leads to reduced future health system cost, and positive social and economic consequences for a TB patient.” (p. 14-15, lines 295-297). 

“Four of the five upper middle-income countries (80%) included in this study were among those who neither agreed nor disagreed.” (p. 15, lines 315-316)

“…both estimates included NTP managers from low-, lower middle- and upper middle-income countries.” (p. 15, lines 319-320) 

“Three (14%) (two lower middle-income countries and one low-income country) specified targets for the number of people tested.” (p. 16, lines 227-338)

“According to WHO, the NTP budget in five of the included countries does not include any international funding [6].” (p. 17, lines 369-370)

“We investigated if upper middle-income countries used national evidence more compared to other countries, given that many have a better national research infrastructure, but replies were mixed.” (p. 19, lines 416-419)

“The lack of resources is also reflected in the World TB Report 2019 [6], which showed that only five of the high TB burden countries had a national TB budget consisting of more than 50% domestic resources. Another five countries had TB budgets largely based on international funding, while the budget of 11 countries remained mostly unfunded [6].” (p. 24, lines 520-524)

REVIEWER 2 COMMENT

27. There are countries with very specific ACF (and contact investigation) policies, but due to resource and infrastructural constraints, actual programs may not be adequately operating. Do responses from these NTP mangers (their understanding of existence of these policies and programmatic limitations) match what’s been reported in the public (via policy documents, WHO statements, etc.)? 

AUTHOR RESPONSE

The responses from NTP managers with regards resource constraints do match what has been reported in some National TB Strategic Plans, e.g. p. 17, lines 357-359: “Focusing on human resources, many NSPs documented the need to not only expand human resources, but to increase their capacity for ACF, and to improve their coordination, engagement and supervision.”. 

Moreover, while we have reported the NTP managers qualitative answers on resource constraints in the Results section, Appendix 1 “Data extraction table” provides further insights into how the NSPs have reflected operational considerations, e.g. health system requirements and other operational consideration (columns Q-T). 

We agree that it would be very interesting and valuable to explore whether NTP managers’ responses match what has been reported in public (beyond the NSPs) and hope that future studies may address this important issue.

REVIEWER 2 COMMENT

28. The authors also state in their limitations (Lines 568 - 575) that participant responses may be affected by the mood state bias. Can the authors provide which of the survey questions may be most affected by this bias and how future research (e.g. if the authors were to conduct this study again, what would they do to reduce this bias?) can improve upon this issue? 

AUTHOR RESPONSE 

We have added some information on the questions which we think may have been more affected by mood state bias than others: “It is possible that questions about the use of evidence may have been affected by this bias, as NTP managers who participated in the survey while at a WHO meeting may have been exposed to WHO guidelines more recently and thus valued them more in their responses.” (p. 30, lines 681-684)

In cases where mood state bias may have been caused by the location of the interview, future research may improve upon this issue by conducting interviews under similar conditions, as much as possible. Meanwhile, we acknowledge that making participation convenient for respondents required us to conduct the interviews in different settings and that the participation rate may have otherwise dropped considerably. In the case of NTP managers or other high-level officials, we believe it may be better to focus on convenience of participation and accept risk of mood state bias. 

REVIEWER 2 COMMENT

29. Also, it would be important to mention in the study limitations (or anywhere else in the discussion section) that policy/implementation/ scale-up issues for ACF may be unique to situations (political, financial, and infrastructural) faced in each country. Therefore, similar types of (improved to include additional questions relevant to the country/setting) survey administered in each country to identify challenges and potential strategies in developing policies, implementing and scaling up ACF interventions would be very important. It would be useful for the authors to mention this as part of their future direction for research. 

AUTHOR RESPONSE

We have reflected this in the Future research section, as we fully agree that developing a better contextual understanding for ACF policy development, implementation and scale-up will be of utmost importance moving forward: “Similar types of surveys and mixed-methods studies targeting sub-national stakeholders may provide relevant local evidence for ACF policy development, implementation and scale-up, complementing the findings of this study.” (p. 29, lines 649-652)

REVIEWER 2 COMMENT

30. Based on reporting of the survey results in Figures S3 and 4, this reviewer has one major concern for the survey design. These findings would be been much more useful (again this is my personal opinion and other reviewers may have different perspectives on this matter) if the authors asked the NTP managers to rank each component for the risk factors for ACF and use of evidence in policy making for ACF. For example, it looks clear from Figure S3 that most of the NTP managers feel that increased health systems cost (short-term) is the main perceived risk for ACF intervention, followed by ‘increased worry about health’ and ‘stigma’. Current presentation of survey results in these two figures are a bit difficult to comprehend. 

AUTHOR RESPONSE 

To make Figures 3 and 4 more comprehensive, we have changed the order of the bars aiming to show which aspects NTP managers agreed with most to least. We furthermore agree that future surveys could change the design of these questions to rankings rather than ratings. However, it may be important to note that based on our experience of having included one ranking question (2.4 “How would you rank the importance of ACF among other TB interventions for early case detection? Please rank from 1-7 [1 = most important]), we felt that this type of question seemed considerably more difficult and less intuitive for respondents to answer. In fact, many respondents chose to skip this questions as it took too long consider. This is why data on this question remained incomplete and we did not include it in the final analysis. 

REVIEWER 2 COMMENT

31. Figure S1. Please provide sub-titles describing each of the process a bit more in detail so that readers can have a good understanding just by reviewing the figure and the sub-title itself.

AUTHOR RESPONSE

We have provided sub-titles as suggested. For Figure 1: “23 National Tuberculosis (TB) Programme managers participated in the cross-sectional survey, which was analyzed using descriptive statistics and manifest content analysis. 22 National TB Strategic Plans were reviewed, and data extracted to complement the survey findings. All data were analyzed in parallel and merged in the interpretation of the findings.” (p. 7, text box, line 30) 

REVIEWER 2 COMMENT

32. Table 1: It would be good to have a full version of such table presented in the appendix to show the process of developing the survey (this would be very helpful for other researchers to use this study in developing similar types of survey for other disease or for TB within the country) 

AUTHOR RESPONSE

We think that the survey questionnaire (Appendix 2) may be most helpful for other researchers to use for developing similar types of surveys for other diseases or for TB within the country.

As the full qualitative analysis table contains meaning units which are direct quotes from the open-ended questions and may compromise NTP managers anonymity, we are, unfortunately, unable to share the table as ethical restrictions apply. 

REVIEWER 2 COMMENT

33. In Figure S2 (Policy analysis triangle), it would be very important to provide sub-text/title for each content of the triangle. For example, it’s not clear from just reviewing the figure itself how these components interact with one another. To this reviewer, a policy triangle is better represented as a process diagram showing how each component interacts with one another and describing the contents of each triangle. 

AUTHOR RESPONSE

We have provided sub-titles for Figure 2: “This study showed that the content of active case-finding (ACF) policies, including their development, implementation and scale-up, is influenced by context, actors and processes. Example topics: Context – Important factors included the perceptions of the benefits and risks of ACF and the availability of resources. Actors – Key actors comprised the national government, managers in districts and regions and non-governmental organizations (NGOs). Process – Important processes were evidence use and institutionalized processes to facilitate the same, e.g. regularly convening working groups.” (p. 8, text box, line 144)

We have constructed Figure 2 to give an insight into factors that play a role in ACF policy development, implementation and scale-up across the high TB burden countries, highlighting a few examples. Yet, we strongly agree that the interaction between the different factors that influence ACF policy development, implementation and scale-up would be important to investigate. Such an analysis may provide an in-depth understanding of these processes in a given context and we hope future studies may address this.

---

## [Decision Letter · Decision Letter 2]

1 Oct 2020

Active case-finding policy development, implementation and scale-up in high-burden countries: a mixed-methods survey with National Tuberculosis Programme managers and document review

PONE-D-20-12519R2

Dear Ms Biermann,

We’re pleased to inform you that your manuscript has been judged scientifically suitable for publication and will be formally accepted for publication once it meets all outstanding technical requirements.

Kind regards,

Hemant Deepak Shewade, MBBS MD

Academic Editor

PLOS ONE

Additional Editor Comments (optional):

Reviewers' comments:

Reviewer's Responses to Questions

**Comments to the Author**

1. If the authors have adequately addressed your comments raised in a previous round of review and you feel that this manuscript is now acceptable for publication, you may indicate that here to bypass the “Comments to the Author” section, enter your conflict of interest statement in the “Confidential to Editor” section, and submit your "Accept" recommendation.

Reviewer #1: All comments have been addressed

2. Is the manuscript technically sound, and do the data support the conclusions?

Reviewer #1: Yes

3. Has the statistical analysis been performed appropriately and rigorously? 

Reviewer #1: Yes

4. Have the authors made all data underlying the findings in their manuscript fully available?

Reviewer #1: Yes

5. Is the manuscript presented in an intelligible fashion and written in standard English?

Reviewer #1: Yes

6. Review Comments to the Author

Reviewer #1: The evidence generated through this research would enable the policy makers of NTPs to reflect on the process they follow in making decisions.

7. PLOS authors have the option to publish the peer review history of their article (what does this mean?). If published, this will include your full peer review and any attached files.

Reviewer #1: **Yes: **Pruthu Thekkur

---

## [Editor Report · Acceptance letter]

2 Oct 2020

PONE-D-20-12519R2 

Active case-finding policy development, implementation and scale-up in high-burden countries: a mixed-methods survey with National Tuberculosis Programme managers and document review 

Dear Dr. Biermann:

I'm pleased to inform you that your manuscript has been deemed suitable for publication in PLOS ONE. Congratulations! Your manuscript is now with our production department. 

Kind regards, 

on behalf of

Dr. Hemant Deepak Shewade 

Academic Editor

PLOS ONE